

# Avaframe com1DFA (version 1.3): a thickness integrated computational avalanche module - Theory, numerics and testing

Matthias Tonnel[1], Anna Wirbel[1], Felix Oesterle[1], and Jan-Thomas Fischer[1]

[1]Dep. of Natural Hazards, BFW - Austrian Research Center for Forests, Innsbruck, Austria

**Correspondence:** Felix Oesterle (felix@avaframe.org)

**Abstract.** Simulation tools are important to investigate and predict mobility and the destructive potential of gravitational mass flows (e.g. snow avalanches). AvaFrame - the open avalanche framework - offers well established computational modeling approaches, tools for data handling and analysis as well as ready to use modules for evaluation and testing. This paper presents the theoretical background, derivation and model verification for one of AvaFrame's core modules, the thickness integrated

computational model for dense flow avalanches, named com1DFA. Particular emphasis within the description of the utilized numerical particle grid method is given to the computation of spatial gradients and the accurate implementation of driving and resisting forces. The implemented method allows to provide a time-space criterion connecting the numerical particles, grid and time discretization. The convergence and robustness of the numerical implementation is checked with respect to the spatio-temporal evolution of the flow variables using tests with a known analytical solution. In addition we present a new

test for verifying the accuracy of the numerical simulation in terms of runout (angle and distance). This test is derived from the total energy balance along the center of mass path of the avalanche. This manuscript, particularly in combination with the code availability (open-source code repository) and detailed online documentation provides a description of an extendable framework for modeling and verification of avalanche simulation tools.

## 1 Introduction

Simulation tools for gravitational mass flows - with a focus on snow avalanches in this manuscript - are in great demand for operational engineering applications, scientific model development and gain increasing attention in academic education. Each of these application requires different outputs. For operational engineering applications, the runout outcome for different scenarios is usually of highest interest. Scientific applications aim at better understanding the processes and will require outputs such as flow variable evolution. Existing tools for simulating snow avalanches cover a wide range of numerical implementations and

vary from proprietary (e.g. Christen et al., 2010; Sampl and Zwinger, 2004; Zugliani and Rosatti, 2021; Li et al., 2021) to open source software (e.g. Hergarten and Robl, 2015; Mergili et al., 2017; Rauter et al., 2018). The latter ones are generally more focused towards scientific and academic issues whereas the first are more geared towards operational applications. AvaFrame - the open avalanche framework - strives to fill the gap between operations and scientific development combining over a decade of operational application (Sampl and Zwinger, 2004; Fischer et al., 2014) with an open source scientific development envi-

ronment. Using a modular structure AvaFrame adds in-depth testing and analysis modules to the core flow modules. Further



modules provide interfaces for visualization and geodata handling for all kinds of existing and emerging simulation tools. It enables to combine, further develop and extend the different tools to best suit the users needs.

At their core avalanche simulation tools are based on a large variety of flow models, differing by their basic assumptions (what physical processes they include, degree of complexity), mathematical derivation and/or their numerical implementation.

These range from Eulerian methods (Christen et al., 2010; Mergili et al., 2017; Rauter and Tuković, 2018; Zugliani and Rosatti, 2021) using spatially fixed meshes to Lagrangian methods (Sampl and Zwinger, 2004) where the mass is discretized. Some methods are combinations of Eulerian and Lagrangian approaches such as the Material Point method (Stomakhin et al., 2013). AvaFrame's **com1DFA** DFA module is based on a flow model that is derived from the thickness integration of the conservation equations of mass and momentum. Classic shallow water models, e.g. Saint-Venant, integrate in the direction of gravity, often

called "depth" integration. Other approaches, such as Savage Hutter models (Savage and Hutter, 1989) integrate in the slope normal direction but also call it "depth" integrated. To be more consistent with operational terminology, we propose to call the integration in slope normal direction "thickness" integration. This way, we can highlight the special case of gravitational mass flows in steep terrain .

The resulting equations are solved using a mixed particle grid method in which mass is tracked using particles. Pressure

gradients are computed using a SPH method adapted to steep terrain and flow thickness is computed on the grid (Sampl and Zwinger, 2004; Monaghan, 2005; Liu and Liu, 2010). To avoid nonphysical behavior in regards to starting and stopping, **com1DFA** applies the method proposed in (Mangeney-Castelnau et al., 2003), allowing for a friction balanced starting and stopping of the flow.

Verifying and validating the methods applied in our implementation is a challenging but crucial step, as it is for all simulation

tools. Verification is done by comparing the numerical model results to an analytical solution (e.g. Zugliani and Rosatti, 2021; Rauter and Tuković, 2018). Validation can be tackled in different ways, either by comparing the model results to observations (e.g. Christen et al., 2010) or by comparing them to the results of already trusted numerical models (e.g. Zugliani and Rosatti, 2021). In this manuscript, the focus is given to model verification and the numerical model results are compared to tests with known (semi-) analytical solutions using two different approaches.

The flow variable tests (Hutter et al., 1993; Faccanoni and Mangeney, 2013) allow us to investigate the local spatio-temporal evolution of flow thickness and velocity. This enables to verify the proper implementation of the pressure gradient and friction force computation, among others. In contrast, the energy line test is based on the investigation of the total energy balance (Körner, 1980). It focuses on the accuracy of the global kinetic energy (velocity) along the path and the corresponding center of mass runout. Thereby the proper implementation of stopping behavior, that is to say the proper balancing of driving and

friction forces, can be assessed. And provides a test where the runout, a quantity which is important for operational applications, is tested. We explore and explain the limitations of these two approaches.

The manuscript itself is structured as follows: Section 2 summarizes the underlying flow model, including fundamental assumptions and derivations of the thickness integrated equations, building the foundation for the gravitational mass flow simulations. In Sect. 3 the spatial and temporal discretizations of the model equations are described using a particle grid

approach. The implementation into the AvaFrame computational module **com1DFA** is outlined in Sect. 4. Model verification





tests are presented in Sect. 5, targeting the correct implementation of the mathematical model as well as the convergence and robustness of the numerical model code. In addition to employing test cases with a known (semi-)analytical solution, a new energy line test is introduced in Sect. 5.2.

Besides the in depth description within this manuscript AvaFrame, Oesterle et al. (2022) provides a combination of code and
corresponding documentation. Users find more information according to their individual scientific, operational or educational focus and the reader is invited to contribute to the future development. It is important to note that this manuscript presents the latest development state of the **com1DFA** module. It differs slightly from the implementation of the **com1DFA** module used for operational purposes which is described in the online AvaFrame documentation. For example differences include improvements of the SPH gradient computation method and how friction forces are taken into account.

## 2  Mathematical model: From 3D equations to thickness integrated Lagrangian equations

In this section, the mathematical model and associated equations used to simulate DFA processes are presented. The derivation is based on the thickness integration of the three-dimensional Navier-Stokes equations, using a Lagrangian approach with a terrain following coordinate system. The equations are simplified using the assumption of shallow flow on a mildly curved topography, meaning flow thickness is considerably smaller than the length and width of the avalanche and it is considerably
smaller than the topography curvature radius.

We consider snow as the material, however the choice of material does not influence the derivation in the first place. We assume constant density $\rho_0$ and a flow on a surface $\mathcal{S}$, subjected to the gravity force and friction on the surface $\mathcal{S}$. If needed, additional processes such as entrainment or other external effects can be taken into account. These processes are included in the following derivations but will not be considered for model verification (Sect. 5), as test cases with an analytical solution
are only available for simplified conditions where entrainment or any additional forces are discarded. The mass conservation equation applied to a Lagrangian volume of material $V(t)$ reads:

$$\frac{\mathrm{d}}{\mathrm{d}t} \underbrace{\int_{V(t)} \rho_0 \, \mathrm{d}V}_{m(t)} = \rho_0 \frac{\mathrm{d}V(t)}{\mathrm{d}t} = \oint_{\partial V(t)} q^{\mathrm{ent}} \, \mathrm{d}A, \qquad (1)$$

where the source term $q^{\mathrm{ent}}$ represents the snow entrainment rate (incoming mass flux) and the momentum conservation equation applied to same volume of material $V(t)$ reads:

$$\frac{\mathrm{d}}{\mathrm{d}t} \int_{V(t)} \rho_0 \, \mathbf{u} \, \mathrm{d}V = \underbrace{\oint_{\partial V(t)} \boldsymbol{\sigma} \cdot \mathbf{n} \, \mathrm{d}A}_{\text{surface forces}} + \underbrace{\int_{V(t)} \rho_0 \, \mathbf{g} \, \mathrm{d}V}_{\text{body force}} + \mathbf{F}^{\mathrm{ext}}, \qquad (2)$$

where $\mathbf{u}$ is the fluid velocity and $\mathbf{g}$ the gravity acceleration. $\boldsymbol{\sigma} = -p\mathbb{1} + \mathbf{T}$ represents the stress tensor, where $\mathbb{1}$ is the identity tensor, $p$ the pressure and $\mathbf{T}$ the deviatoric part of the stress tensor. $\mathbf{n}$ is the normal vector to $\partial V(t)$. $F^{\mathrm{ext}}$ represent additional



forces due to snow entrainment (force needed to break and compact the entrained snow) or due to added resistance (trees, obstacles, etc.). $F^{\text{ext}}$ is assumed to be surface parallel. Problem-specific assumptions are needed to simplify the mass and momentum conservation equations. These are introduced in the following sections alongside a local coordinate system, boundary conditions and constitutive relations.

## 2.1  Natural Coordinate System (NCS) and thickness integrated quantities

**Hypothesis**: *A control volume $V(t)$ is assumed to be a small prism shape extending from the bottom surface $\mathcal{S}_b$ (lying on the topography $\mathcal{S}$) up to the free surface in the surface normal direction $\mathbf{N^b}$ as illustrated in Fig. 1. Note that the bottom surface $\mathcal{S}_b$ of area $A^b$ has no predefined shape. The octogonal shape used in Fig. 1 is just one possible option.*

The normal vector $\mathbf{N^b}$ to the bottom surface is pointing upwards whereas $\mathbf{n^b} = -\mathbf{N^b}$ is the bottom normal vector to the Lagrangian control volume (pointing out of the volume).

$$V(t) = \int\limits_{V(t)} \mathrm{d}V = \int\limits_{\mathcal{S}_b} \left( \int\limits_b^s \det(\mathbf{J}) \, \mathrm{d}x_3 \right) \mathrm{d}A \tag{3}$$

where $\mathbf{J}$ is the transformation matrix from the Cartesian coordinate system to the Natural coordinate system (NCS). The NCS is an orthonormal coordinate system $(\mathbf{v_1}, \mathbf{v_2}, \mathbf{v_3})$ aligned with the bottom surface. $\mathbf{v_3} = \mathbf{N^b} = -\mathbf{n^b}$ is the normal vector to the bottom surface pointing upwards. $\mathbf{v_1}$ is pointing in the direction of the thickness integrated fluid velocity $\overline{\mathbf{u}}$ (introduced below).

$$\mathbf{v_1} = \frac{\overline{\mathbf{u}}}{\|\overline{\mathbf{u}}\|}, \quad \mathbf{v_2} = \frac{\mathbf{v_3} \wedge \mathbf{v_1}}{\|\mathbf{v_3} \wedge \mathbf{v_1}\|}, \quad \mathbf{v_3} = \mathbf{N^b} \tag{4}$$

In the case of shallow flow on moderately curved surfaces, $\det(\mathbf{J}) = (1 - \kappa_1 x_3)(1 - \kappa_2 x_3) \approx 1$. $\kappa_{\{1,2\}}$ represent the surface curvature in $\mathbf{v}_{\{1,2\}}$ directions and $x_3$ is the elevation from the bottom surface in the direction $\mathbf{N^b}$. This approximation is valid if the curvature radius is much larger than the flow thickness $h$. In this case, the control volume reads:

$$V(t) \approx \int\limits_{\mathcal{S}_b} \underbrace{\int\limits_b^s \mathrm{d}x_3}_{h(t)} \mathrm{d}A \tag{5}$$

The following volume (indicated by the superscript $\widetilde{\square}$), area (indicated by the superscript $\widehat{\square}$) and thickness (indicated by the superscript $\overline{\square}$) averaged quantities are introduced (where $f$ is a scalar or vector function of $\Omega$):

$$\widetilde{f} = \frac{1}{V(t)} \int\limits_{V(t)} f \, \mathrm{d}V$$

$$\widehat{f} = \frac{1}{A^b(t)} \int\limits_{\mathcal{S}_b} f \, \mathrm{d}A \quad \text{and}$$

$$\overline{f} = \frac{1}{h(t)} \int\limits_0^{h(t)} f \, \mathrm{d}x_3$$

$$\widetilde{f}(x_3) = \frac{1}{A^b(\widehat{h} - x_3)} \int\limits_{\mathcal{S}_b} \left( \int\limits_{x_3}^{h(t)} f \det(\mathbf{J}) \, \mathrm{d}x_3 \right) \mathrm{d}A \approx \frac{1}{A^b(\widehat{h} - x_3)} \int\limits_{\mathcal{S}_b} \left( \int\limits_{x_3}^{h(t)} f \, \mathrm{d}x_3 \right) \mathrm{d}A$$

$$\overline{f}(x_3) = \frac{1}{(h - x_3)} \int\limits_{x_3}^{h(t)} f \, \mathrm{d}x_3. \tag{6}$$





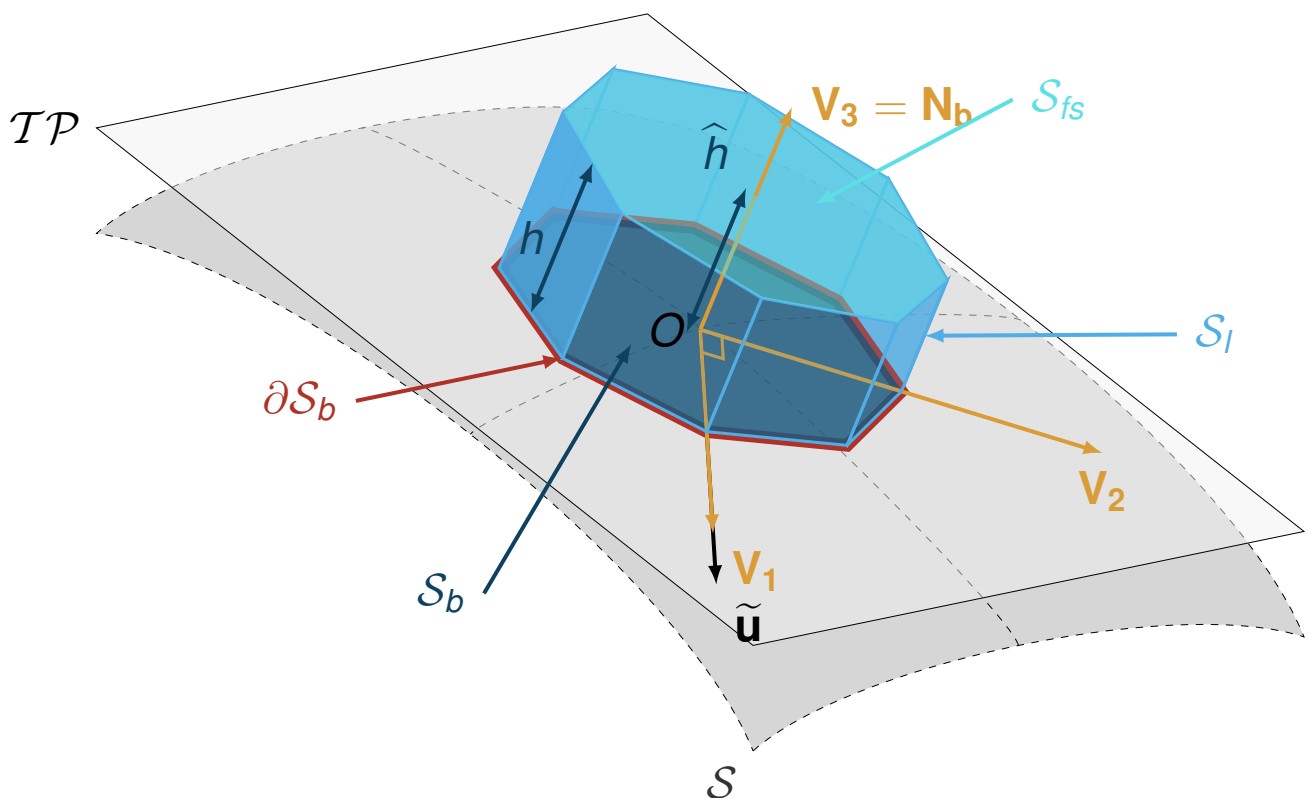

**Figure 1.** Example of small Lagrange volume considered in the equations and corresponding notations. The gray surface (denoted $\mathcal{S}$) represents the bottom surface (topography) and $\mathcal{TP}$ represents the tangent plane to the surface at the the point O. The normal vector to $\mathcal{S}$ and $\mathcal{TP}$ in O is $\mathbf{v_3} = \mathbf{N^b}$. The control volume, represented in blue, has a basal surface $\mathcal{S}_b$ lying in $\mathcal{TP}$, a lateral surface $\mathcal{S}_l$ aligned with $\mathbf{N^b}$ and a free surface $\mathcal{S}_{fs}$.

Note that $\widetilde{f}(0) = \widetilde{f}$ and $\overline{f}(0) = \overline{f}$. When the control volume goes to 0, i.e. basal area goes to a point, $\widetilde{f} \xrightarrow{A^b \to 0} \overline{f}$ and $\widehat{f} \xrightarrow{A^b \to 0} f$.

The NCS has some interesting properties that will be useful for projecting and solving the equations. Because of the orthogonality of this NCS, we have $\mathbf{v}_i \cdot \mathbf{v}_j = \delta_{ij}, \{i,j\} \in \{1,2,3\}^2$ which gives after time derivation:

$$\frac{\mathrm{d}(\mathbf{v}_i \cdot \mathbf{v}_j)}{\mathrm{d}t} = \mathbf{v}_i \cdot \frac{\mathrm{d}\mathbf{v}_j}{\mathrm{d}t} + \frac{\mathrm{d}\mathbf{v}_i}{\mathrm{d}t} \cdot \mathbf{v}_j = 0, \tag{7}$$

meaning that:

$$\begin{cases} \dfrac{\mathrm{d}\mathbf{v}_i}{\mathrm{d}t} \cdot \mathbf{v}_i = 0 \implies \dfrac{\mathrm{d}\mathbf{v}_i}{\mathrm{d}t} \perp \mathbf{v}_i \\ \dfrac{\mathrm{d}\mathbf{v}_i}{\mathrm{d}t} \cdot \mathbf{v}_j = -\mathbf{v}_i \cdot \dfrac{\mathrm{d}\mathbf{v}_j}{\mathrm{d}t}, \quad i \neq j. \end{cases} \tag{8}$$



It is possible to express $\frac{d\mathbf{v}_1}{dt}$ in terms of $(\mathbf{v}_1, \mathbf{v}_2, \mathbf{v}_3)$ and using orthogonality of $\frac{d\mathbf{v}_i}{dt}$ and $\mathbf{v}_i$:

$$\frac{d\mathbf{v}_1}{dt} = \alpha_i \mathbf{v}_i = \alpha_1 \mathbf{v}_1^{\ \mathbf{0}} + \alpha_2 \mathbf{v}_2 + \alpha_3 \mathbf{v}_3, \quad \alpha_i = \frac{d\mathbf{v}_1}{dt} \cdot \mathbf{v}_i \tag{9}$$

The derivative of the thickness integrated velocity decomposes to:

$$\frac{d\overline{\mathbf{u}}}{dt} = \frac{d(\overline{u}_1 \mathbf{v}_1)}{dt} = \overline{u}_1 \frac{d\mathbf{v}_1}{dt} + \frac{d\overline{u}_1}{dt}\mathbf{v}_1 = \overline{u}_1(\alpha_2 \mathbf{v}_2 + \alpha_3 \mathbf{v}_3) + \frac{d\overline{u}_1}{dt}\mathbf{v}_1 \tag{10}$$

### 2.2   Boundary conditions

To complete the conservation Eqs. 1 and 2 the following boundary conditions at the bottom ($\mathcal{S}_b$) and free ($\mathcal{S}_{fs}$) surfaces are introduced. $\sigma_s$, respectively $\sigma_b$, represent the restriction of $\sigma$ to the free surface $\mathcal{S}_{fs}$, respectively the bottom surface $\mathcal{S}_b$:

125        – traction free free-surface:

$$\boldsymbol{\sigma_s} \cdot \mathbf{n_s} = \mathbf{0} \quad \text{on } \mathcal{S}_{fs} \tag{11}$$

– impenetrable bottom surface without detachment:

$$\mathbf{u^b} \cdot \mathbf{n^b} = \mathbf{0} \quad \text{on } \mathcal{S}_b \tag{12}$$

– bottom friction law:

$$\boldsymbol{\tau^b} = \boldsymbol{\sigma^b} \cdot \mathbf{n^b} - ((\boldsymbol{\sigma^b} \cdot \mathbf{n^b}) \cdot \mathbf{n^b})\mathbf{n^b} = \mathbf{f}(\boldsymbol{\sigma^b}, \overline{\mathbf{u}}, h, \rho_0, t, \mathbf{x}) = -f(\boldsymbol{\sigma^b}, \overline{\mathbf{u}}, h, \rho_0, t, \mathbf{x})\mathbf{v}_1 \quad \text{on } \mathcal{S}_b \tag{13}$$

### 2.3   Constitutive relation: friction force

Up to now in the derivations, the bottom shear stress $\boldsymbol{\tau^b}$ is unknown. To close the momentum equation, a constitutive equation describing the basal shear stress tensor $\boldsymbol{\tau^b}$ as a function of the avalanche flow state is required:

$$\boldsymbol{\tau^b} = \mathbf{f}(\boldsymbol{\sigma^b}, \overline{\mathbf{u}}, h, \rho_0, t, \mathbf{x}) \tag{14}$$

In the following, we employ a Mohr-Coulomb friction model that describes the friction interaction between two solids. The bottom shear stress reads:

$$\boldsymbol{\tau^b} = -\tan\delta\,\boldsymbol{\sigma^b} \cdot \mathbf{n^b}\,\frac{\overline{\mathbf{u}}}{\|\overline{\mathbf{u}}\|}, \tag{15}$$

where $\delta$ is the friction angle and $\mu = \tan\delta$ is the friction coefficient. The bottom shear stress linearly increases with the normal stress component $p^b$ (Bartelt et al., 1999).

With Mohr-Coulomb friction an avalanche starts to flow if the slope inclination exceeds the friction angle $\delta$. In the case of an infinite slope of constant inclination, the avalanche velocity would increase indefinitely. Using this friction law, flow velocity





is overestimated and hence is not suited to model the flow of snow avalanches. However, because of its relative simplicity, the Mohr-Coulomb friction model is convenient for deriving analytical solutions and testing numerical implementations.

Different friction models accounting for the influence of flow velocity, flow thickness, etc. have been proposed (e.g. the Voellmy model Voellmy, 1955). Changing the friction model means changing the $\mathbf{f}$ function (Eq. 14). In the **com1DFA** module, three friction models are available. First a Coulomb one which is used in this paper. Second a Voellmy friction model (Voellmy, 1955) and third the samosAT friction model which is the one used for hazard mapping by Austrian federal agencies (Sampl, 2007).

### 2.4 Expression of surface forces in the NCS

Taking advantage of the NCS and using the boundary conditions, it is possible to split the surface forces into bottom, lateral and free surface forces and perform further simplifications:

$$\oint_{\partial V(t)} \boldsymbol{\sigma} \cdot \mathbf{n}\,\mathrm{d}A = \int_{\mathcal{S}_b} \boldsymbol{\sigma^b} \cdot \mathbf{n^b}\,\mathrm{d}A + \int_{\mathcal{S}_{fs}} \boldsymbol{\sigma_s} \cdot \overset{\mathbf{0}}{\mathbf{n_s}}\,\mathrm{d}A + \int_{\mathcal{S}_{lat}} \boldsymbol{\sigma} \cdot \mathbf{n}\,\mathrm{d}A = \underbrace{\int_{\mathcal{S}_b} \boldsymbol{\sigma^b} \cdot \mathbf{n^b}\,\mathrm{d}A}_{\text{bottom force}} + \underbrace{\oint_{\partial \mathcal{S}_b} \left( \int_0^h \boldsymbol{\sigma} \cdot \mathbf{n}\,\mathrm{d}x_3 \right)\mathrm{d}l}_{\text{lateral force}}. \tag{16}$$

Using the notations introduced in Sect. 2.1 and the decomposition of the stress tensor, the bottom force can be expressed as a surface normal component and a surface tangential one:

$$\int_{\mathcal{S}_b} \boldsymbol{\sigma^b} \cdot \mathbf{n^b}\,\mathrm{d}A = \int_{\mathcal{S}_b} (-p^b \mathbb{1} + \mathbf{T}) \cdot \mathbf{n^b}\,\mathrm{d}A = -\int_{\mathcal{S}_b} p^b \mathbf{n^b}\,\mathrm{d}A + \int_{\mathcal{S}_b} \mathbf{T} \cdot \mathbf{n^b}\,\mathrm{d}A = -\int_{\mathcal{S}_b} p^b \mathbf{n^b}\,\mathrm{d}A + \int_{\mathcal{S}_b} \boldsymbol{\tau^b}\,\mathrm{d}A = -A^b \widehat{p^b \mathbf{n^b}} + A^b \widehat{\boldsymbol{\tau^b}}, \tag{17}$$

where $\boldsymbol{\tau^b}$ is the basal friction term (introduced in Sect. 2.2). Applying Green's theorem, the lateral force reads:

$$\oint_{\partial \mathcal{S}_b} \left( \int_0^h \boldsymbol{\sigma} \cdot \mathbf{n}\,\mathrm{d}x_3 \right)\mathrm{d}l = \oint_{\partial \mathcal{S}_b} \left( \int_0^h (-p\mathbb{1} + \mathbf{T})\,\mathrm{d}x_3 \right) \cdot \mathbf{n}\,\mathrm{d}l = -\oint_{\partial \mathcal{S}_b} \left( \int_0^h p\,\mathrm{d}x_3 \right) \cdot \mathbf{n}\,\mathrm{d}l + \oint_{\partial \mathcal{S}_b} \left( \int_0^h \mathbf{T}\,\mathrm{d}x_3 \right) \cdot \mathbf{n}\,\mathrm{d}l$$
$$= -\oint_{\partial \mathcal{S}_b} h\overline{p}\mathbf{n}\,\mathrm{d}l + \oint_{\partial \mathcal{S}_b} h\overline{\mathbf{T}} \cdot \mathbf{n}\,\mathrm{d}l = -\int_{\mathcal{S}_b} \boldsymbol{\nabla} h\overline{p}\,\mathrm{d}A + \int_{\mathcal{S}_b} \boldsymbol{\nabla} \cdot h\overline{\mathbf{T}}\,\mathrm{d}A = -A^b \widehat{\boldsymbol{\nabla} h\overline{p}} + A^b \widehat{\boldsymbol{\nabla} \cdot h\overline{\mathbf{T}}} \tag{18}$$

Equations 17 and 18 represent the thickness integrated form of the surface forces and can now be used to write the thickness integrated momentum equation.

### 2.5 Thickness integrated momentum equation

Using the definitions of average values given in Sect. 2.1 and the decomposition of the surface forces given by Eqs. 17 and 18 combined with the expression of the entrainment force detailed in Appendix A, the momentum equation reads:

$$\rho_0 \frac{\mathrm{d}(V(t)\widetilde{\mathbf{u}})}{\mathrm{d}t} = \rho_0 V \frac{\mathrm{d}\widetilde{\mathbf{u}}}{\mathrm{d}t} + \rho_0 \widetilde{\mathbf{u}} \frac{\mathrm{d}V}{\mathrm{d}t} = \oint_{\partial V(t)} \boldsymbol{\sigma} \cdot \mathbf{n}\,\mathrm{d}A + \rho_0 V \mathbf{g} + \mathbf{F}^{\text{ext}} \tag{19}$$



which leads to:

$$
\quad \rho_0 V \frac{\mathrm{d}\widetilde{\mathbf{u}}}{\mathrm{d}t} = \underbrace{-A^b \widehat{p\mathbf{n}^{\mathbf{b}}}}_{\substack{\text{bottom} \\ \text{normal force}}} \underbrace{+A^b \widehat{\boldsymbol{\tau}^{\mathbf{b}}}}_{\substack{\text{bottom} \\ \text{shear force}}} \underbrace{-A^b \widehat{\boldsymbol{\nabla} h \overline{p}}}_{\substack{\text{lateral} \\ \text{pressure force}}} \overbrace{\underbrace{+A^b \widehat{\boldsymbol{\nabla} \cdot h \overline{\mathbf{T}}}}_{\substack{\text{lateral} \\ \text{shear force}}}}^{O(\boldsymbol{\epsilon}^2)} +\rho_0 V \mathbf{g} + \mathbf{F}^{\text{ext}} \underbrace{-\widetilde{\mathbf{u}} \oint_{\partial V(t)} q^{\text{ent}}\, \mathrm{d}A}_{\substack{\text{momentum loss} \\ \text{entrainment}}} \tag{20}
$$

The lateral shear stress term is neglected because of its relative smallness in comparison to the other terms as shown by the dimensional analysis carried out in Gray and Edwards (2014). The mass conservation reads:

$$
\rho_0 \frac{\mathrm{d}V}{\mathrm{d}t} = \oint_{\partial V(t)} q^{\text{ent}}\, \mathrm{d}A, \tag{21}
$$

Using the approximations from Sect. 2.1, the momentum equation becomes:

$$
\quad \rho_0 V \frac{\mathrm{d}\overline{\mathbf{u}}}{\mathrm{d}t} = -A^b p\mathbf{n}^{\mathbf{b}} + A^b \boldsymbol{\tau}^{\mathbf{b}} - A^b \boldsymbol{\nabla} h \overline{p} + \rho_0 V \mathbf{g} + \mathbf{F}^{\text{ext}} - \overline{\mathbf{u}} \oint_{\partial V(t)} q^{\text{ent}}\, \mathrm{d}A, \tag{22}
$$

where all quantities are evaluated at the center of the basal area (point O in Fig. 1). This equation is projected in the normal direction $\mathbf{v_3} = \mathbf{N^b}$ to get the expression of the basal pressure $p^b$. The projection of this same equation on the tangential plane leads to the differential equations satisfied by $\overline{\mathbf{u}}$.

### 2.5.1 Pressure distribution, thickness integrated pressure and pressure gradient

We can project the momentum equation (Eq. 22), using the volume between $x_3$ and the surface $h$, in the normal direction $(\mathbf{v_3} = \mathbf{N^b} = -\mathbf{n^b})$. Applying the properties of the NCS (Eq. 10) the surface normal component of Eq. 22 reads:

$$
\begin{aligned}
\rho_0 V(x_3,t) \frac{\mathrm{d}\overline{\mathbf{u}}(x_3)}{\mathrm{d}t} \cdot \mathbf{v_3} =\ & \rho_0 A^b (h-x_3) \overline{u}_1(x_3) \frac{\mathrm{d}\mathbf{v_1}}{\mathrm{d}t} \cdot \mathbf{v_3} \\
=\ & -\rho_0 A^b (h-x_3) \overline{u}_1(x_3) \mathbf{v_1} \cdot \frac{\mathrm{d}\mathbf{v_3}}{\mathrm{d}t} = -\rho_0 A^b (h-x_3) \overline{\mathbf{u}}(x_3) \cdot \frac{\mathrm{d}\mathbf{N^b}}{\mathrm{d}t} \\
=\ & -A^b p \underbrace{\mathbf{n^b} \cdot \mathbf{N^b}}_{-1} + A^b \underbrace{\boldsymbol{\tau}^{\mathbf{b}} \cdot \mathbf{N^b}}_{0} - A^b \boldsymbol{\nabla}\{(h-x_3)\overline{p}\} \cdot \mathbf{N^b} \\
& + \rho_0 V \underbrace{\mathbf{g} \cdot \mathbf{N^b}}_{g_{N^b}} + \underbrace{\mathbf{F}^{\text{ext}} \cdot \mathbf{N^b}}_{0} - \overline{\mathbf{u}} \underbrace{\oint_{\partial V(t)} q^{\text{ent}}\, \mathrm{d}A \cdot \mathbf{N^b}}_{0}
\end{aligned} \tag{23}
$$

Neglecting the normal component of the pressure gradient gives the expression for pressure. Under the condition that $\overline{\mathbf{u}}_1(x_3)$ is independent of $x_3$, pressure follows a linear profile from the bottom surface to the free surface:

$$
\quad p(x_3) = \rho_0 (h-x_3)\left\{-g_{N^b} - \overline{\mathbf{u}} \cdot \frac{\mathrm{d}\mathbf{N^b}}{\mathrm{d}t}\right\} \quad \text{and} \quad p(x_3=0) = p^b = \rho_0 h\left\{-g_{N^b} - \overline{\mathbf{u}} \cdot \frac{\mathrm{d}\mathbf{N^b}}{\mathrm{d}t}\right\} \tag{24}
$$

Note that the bottom pressure should always be positive. A negative pressure is nonphysical and means that the material is not in contact with the bottom surface anymore. This can happen in the case of large velocities on convex topography. If so, the





material should be in a free fall state until it gets back in contact with the topography. A description on how this is handled within the numerical implementation can be found in Sect. 4.3.

Using the previous result of Eq. 24, it is possible to express the thickness integrated pressure $\overline{p}$:

$$h\overline{p} = \int_0^h p(x_3)\,\mathrm{d}x_3 = -\rho_0 \frac{h^2}{2}\left(g_{\mathbf{N^b}} + \overline{\mathbf{u}}\cdot\frac{\mathrm{d}\mathbf{N^b}}{\mathrm{d}t}\right) = -\rho_0\frac{h^2}{2}\,g^{\mathrm{eff}}, \tag{25}$$

where $g^{\mathrm{eff}}$ is the effective normal acceleration acting on the volume, including the normal component of gravity and a curvature component.

The expression of the thickness integrated pressure is used to derive the pressure gradient $\boldsymbol{\nabla} h\overline{p}$. Assuming $g^{\mathrm{eff}}$ to be locally
constant (otherwise $g^{\mathrm{eff}}$ would remain inside the gradient operator), leads to:

$$\boldsymbol{\nabla} h\overline{p} = -\rho_0\,g^{\mathrm{eff}}\,h\,\boldsymbol{\nabla} h. \tag{26}$$

### 2.5.2   Tangential momentum equation

Using the derived expression of the thickness integrated pressure (Eq. 26), we project the momentum balance (Eq. 22) in the tangent plane leading to the following equation:

$$\rho_0 V\left(\frac{\mathrm{d}\overline{\mathbf{u}}}{\mathrm{d}t} - \left(\frac{\mathrm{d}\overline{\mathbf{u}}}{\mathrm{d}t}\cdot\mathbf{v}_3\right)\mathbf{v}_3\right) = A^b\boldsymbol{\tau}^{\boldsymbol{b}} - \rho_0\,g^{\mathrm{eff}}\,hA^b\,\boldsymbol{\nabla}_s h + \rho_0 V\mathbf{g}_s + \mathbf{F}^{\mathrm{ext}} - \overline{\mathbf{u}}\oint_{\partial V(t)} q^{\mathrm{ent}}\,\mathrm{d}A \tag{27}$$

Where $\boldsymbol{\nabla}_s = \boldsymbol{\nabla} - (\boldsymbol{\nabla}\cdot\mathbf{N^b})\mathbf{N^b}$ respectively $\mathbf{g_s} = \mathbf{g} - (\mathbf{g}\cdot\mathbf{N^b})\mathbf{N^b}$ is the tangential component of the gradient operator respectively of the gravity acceleration.

After replacing the velocity derivative component in the normal direction by the expression developed in Eq. 23, Eq. 27 reads:

$$\rho_0 V\frac{\mathrm{d}\overline{\mathbf{u}}}{\mathrm{d}t} = A^b\boldsymbol{\tau}^{\boldsymbol{b}} - \rho_0\,g^{\mathrm{eff}}\,hA^b\,\boldsymbol{\nabla}_s h + \rho_0 V\mathbf{g}_s + \mathbf{F}^{\mathrm{ext}} - \overline{\mathbf{u}}\oint_{\partial V(t)} q^{\mathrm{ent}}\,\mathrm{d}A - \rho_0 V\underbrace{\left(\overline{\mathbf{u}}\cdot\frac{\mathrm{d}\mathbf{v}_3}{\mathrm{d}t}\right)\mathbf{v}_3}_{\text{curvature acceleration}} \tag{28}$$

## 3   Numerical method: particle grid approach

In the previous section, the momentum equation was derived using a Lagrangian approach. In order to solve this set of equations numerically, we employ a mix of particle and grid approaches. We discretize the material into particles and solve the momentum equation for these particles. However, we use a grid to compute several parameters that are required for the computations as
for example surface normal vectors and flow thickness. A theoretical convergence criterion is described in the last section.

### 3.1   Interpolation between particle and grid values

Topography information is usually provided in a raster format which corresponds to a regular rectilinear mesh, from hereon referred to as grid. In order to get information on the surface elevation and normal vectors, the topography information needs to





be interpolated at the particle locations, and this needs to be repeated at each time step since the particles are moving. Similarly, the particle properties such as mass or momentum, which translate into flow thickness and velocity, also need to be interpolated onto the grid. Grid velocity is required to compute the artificial viscosity term, ensuring numerical stability, see Sect. 4.2. Grid flow thickness is used to compute the particle flow thickness which is required for computing the frictional forces. Due to the utilized regular rectilinear mesh a bilinear interpolation method is applied for its simplicity and suitability. It also ensures the conservation of mass or momentum when interpolating from particles to grid and back. The description of the grid and the interpolation method is detailed in the AvaFrame documentation (https://docs.avaframe.org/en/latest/DFAnumerics.html# interpolation).

## 3.2 The particular momentum equation

Discretizing the material into particles (particle quantities are denoted by the subscript $k$, e.g. $m_k = \rho_0 V_k$ is the mass of particle $k$) leads to the following continuity equation:

$$\frac{\mathrm{d}}{\mathrm{d}t} m_k = A_k^{\mathrm{ent}} q^{\mathrm{ent}} \tag{29}$$

By assuming that the Lagrangian control volume $V$ can be represented by a particle, we can derive to the particular momentum equation in the normal direction and in the tangent plane (the entrainment force is detailed in Appendix A):

$$\begin{cases} p_k^b = \rho_0 \, h_k \, g_k^{\mathrm{eff}} \\ m_k \dfrac{\mathrm{d}\overline{\mathbf{u}}_k}{\mathrm{d}t} = A_k^b \boldsymbol{\tau}^{\boldsymbol{b}} - m_k g_k^{\mathrm{eff}} \, \boldsymbol{\nabla}_s h + m_k \mathbf{g}_s + \mathbf{F}_k^{\mathrm{ext}} - \overline{\mathbf{u}}_k A_k^{\mathrm{ent}} q_k^{\mathrm{ent}} - m_k \left( \overline{\mathbf{u}}_k \cdot \dfrac{\mathrm{d}\mathbf{v}_{3,k}}{\mathrm{d}t} \right) \mathbf{v}_{3,k} \end{cases} \tag{30}$$

In this equation (Eq. 30), flow thickness gradient, basal friction and curvature acceleration terms need to be further developed and discretized.

## 3.3 Flow thickness and its gradient

### 3.3.1 Flow thickness gradient computation using SPH

In order to assess the flow thickness gradient, we employ a SPH method (Smoothed Particles Hydrodynamics Method Liu and Liu (2010)), where the gradient is directly derived from the particles and does not require any mesh. In contrast, a mesh method or a MPM (Material Point Method) would directly use a mesh formulation to approximate the gradient or interpolate the particles properties on an underlying mesh and then approximate the gradient of the flow thickness using a mesh formulation.

In theory, a SPH method does not require any mesh to compute the gradient. However, applying this method requires finding neighbor particles. This process can be sped up with the help of an underlying grid, different neighbor search methods are presented in Ihmsen et al. (2014), a "uniform grid method" is used in this paper.

The SPH method is used to express a quantity (the flow thickness in our case) and its gradient at a certain particle location as a weighted sum of its neighbors properties. The principle of the method is well described in Liu and Liu (2010) and the basic





formula reads:

$$f_k \simeq \langle f_k \rangle = \sum_l f_l A_l W_{kl}$$

$$\boldsymbol{\nabla} f_k \simeq \langle \boldsymbol{\nabla} f_k \rangle = -\sum_l f_l A_l \, \boldsymbol{\nabla} W_{kl} \tag{31}$$

Where $W$ represents the SPH-Kernel function (we employ the spiky kernel, see Eq. B2) and the subscript $l$ denotes the neighbor

particles to particle $k$. This kernel function is designed to satisfy the unity condition, be an approximation of the Dirac function and have a compact support domain (Liu and Liu, 2010).

This method is usually either used in a 3D space where particles move freely in this space and where the weighting factor for the summation is the volume of the particle or on a 2D horizontal plane where the weighting factor for the summation is the area of the particle and the gradient is 2D. Here we want to compute the gradient of the flow thickness on a 2D surface (the

topography) that lives in 3D. The method used is analog to the SPH gradient computation on the 2D horizontal plane but the gradient is 3D and tangent to the surface (colinear to the local tangent plane). The theoretical derivation in B2 shows that the SPH computation is equivalent in applying the 2D SPH method in the local tangent plane instead of in the horizontal plane.

### 3.3.2   Flow thickness computation

The particles flow thickness is computed with the help of the grid. The mass of the particles is interpolated onto the grid using

a bilinear interpolation method (described in Sect. 3.1). Then, dividing the mass at the grid cells by the area of the grid cells, while taking the slope of the cell into account, returns the flow thickness field on the grid. This property is interpolated back to the particles which leads to the particle flow thickness property.

We do not compute the flow thickness directly from the particles properties (mass and position) using a SPH method because it induced instabilities. Indeed, the cases where too few neighbors are found, lead to very small flow thickness which becomes

an issue for flow thickness dependent friction laws. Note that using such a SPH method would lead to a fully particular method. But since the flow thickness is only used in some cases for the friction force computation, using a the previously describe grid method should not affect significantly the computation.

### 3.4   Friction force discretization

Expressing the coulomb friction force term in Eq. 30 for a particle reads:

$$A_k^b \boldsymbol{\tau}^{\boldsymbol{b}} = -A_k^b \tan\delta\, p_k^b \mathbf{v}_1 = -A_k^b \tan\delta\, \rho_0 \, h_k \, g_k^{\mathrm{eff}} \, \mathbf{v}_1 = -m_k \tan\delta\, g_k^{\mathrm{eff}} \, \mathbf{v}_1 = -\big\|\mathbf{F}_k^{\mathrm{fric}}\big\|_{\mathrm{max}} \mathbf{v}_1 \tag{32}$$

This relation stands if the particle is moving. The starting and stopping processes satisfy a different equation and are handled differently in the numerical implementation (using the same equation would lead to a non-physical behavior). This is described in more details in Sect. 4.5.





## 3.5 Time discretization

The momentum equation is solved numerically in time using an Euler time scheme. The time derivative of any quantity $f$ is approximated by:

$$\frac{\mathrm{d}f_k}{\mathrm{d}t} \approx \frac{f_k^{n+1} - f_k^n}{\Delta t} \tag{33}$$

where $\Delta t$ represents the time step and $f^n = f(t^n)$, $t^n = n\Delta t$. For the velocity this reads:

$$\frac{\mathrm{d}\overline{\mathbf{u}}_k}{\mathrm{d}t} \approx \frac{\overline{\mathbf{u}}_k^{n+1} - \overline{\mathbf{u}}_k^n}{\Delta t} \tag{34}$$

The position of the particles is then updated using a centered Euler scheme:

$$\mathbf{x}_k^{n+1} = \mathbf{x}_k^n + \frac{\Delta t}{2m}\left(\overline{\mathbf{u}}_k^{n+1} + \overline{\mathbf{u}}_k^n\right) \tag{35}$$

Taking the forces into account is done in two subsequent steps as forces acting on the particles can be sorted into driving forces and friction forces. Friction forces act against the particle motion only affecting the magnitude of the velocity. They can in no case become driving forces. This is why in a first step the velocity is updated with the driving forces before updating in a
second step the velocity magnitude applying the friction force.

## 3.6 Convergence

We are looking for a criterion that relates the properties of the spatial and temporal discretization to ensure convergence of the numerical solution. Simply decreasing the time step and increasing the spatial resolution, by decreasing the grid cell size and kernel radius and increasing the number of particles, does not ensure convergence. The analysis from Moussa and Vila (2000)
carried out on a very similar problem (hyperbolic non linear transport equation with a particle and SPH method) shows that the kernel radius size can not be varied independently from the time step and number of particles. Indeed, they show that the numerical solution converges towards the solution of the equation at the following condition:

$$\begin{cases} r_{\text{part}} \to 0 \\ r_{\text{kernel}} \to 0 \\ \dfrac{r_{\text{part}}^m}{r_{\text{kernel}}^{m+1}} \to 0 \quad m = 2 \end{cases} \quad \text{and} \quad dt \leq C r_{\text{kernel}} \tag{36}$$

Where $r_{\text{part}}$ represents the "size" of a particle , $r_{\text{kernel}}$ represents the SPH kernel radius, $dt$ is the time step and $C$ a constant.
The conditions in Eq. 36 mean that both $r_{\text{part}}$ (particle size) and $r_{\text{kernel}}$ (SPH kernel radius) need to go to zero but also that the particle size needs to go faster to zero than the SPH kernel radius. Finally, the time step needs to go to zero and this at the same rate as $r_{\text{kernel}}$. The particle size can be expressed as a function of the SPH kernel radius:

$$r_{\text{part}} = \left(\frac{A^b}{\pi}\right)^{1/2} = \left(\frac{A_{\text{kernel}}}{\pi n_{\text{ppk}}}\right)^{1/2} = \frac{r_{\text{kernel}}}{n_{\text{ppk}}^{1/2}}, \tag{37}$$





where the particles basal area was assumed to be a circle. Note that this does not affect the results except adding a different
shape factor in front of this expression. $n_{\mathrm{ppk}}$ is the number of particles per kernel radius and defines the density of the particles
when initializing a simulation. Let $n_{\mathrm{ppk}}$ be defined by a reference number of particles per kernel radius $n_{\mathrm{ppk}}^0 > 0$, a reference
kernel radius $r_{\mathrm{kernel}}^0 > 0$ and real exponent $\alpha$:

$$n_{\mathrm{ppk}} = n_{\mathrm{ppk}}^0 \left( \frac{r_{\mathrm{kernel}}}{r_{\mathrm{kernel}}^0} \right)^{\alpha} \tag{38}$$

This leads to a $r_{\mathrm{part}}$:

$$r_{\mathrm{part}} = \left( \frac{r_{\mathrm{kernel}}^0{}^{\alpha}}{n_{\mathrm{ppk}}^0} \right)^{1/2} r_{\mathrm{kernel}}^{1-\alpha/2} \tag{39}$$

Replacing $r_{\mathrm{part}}$ by the previous equation in Eq. 36 leads to the following condition:

$$\frac{r_{\mathrm{kernel}}^0{}^{\alpha}}{n_{\mathrm{ppk}}^0} r_{\mathrm{kernel}}^{-1-\alpha} \to 0 \tag{40}$$

This brings us to the following choice:

$$\begin{cases} dt = C_{\mathrm{time}} r_{\mathrm{kernel}} \\ n_{\mathrm{ppk}} = n_{\mathrm{ppk}}^0 \left( \frac{r_{\mathrm{kernel}}}{r_{\mathrm{kernel}}^0} \right)^{\alpha} \end{cases} \tag{41}$$

Which satisfies the convergence criterion if:

$$\alpha < -1 \tag{42}$$

Note that this criterion leaves some freedom on the choice of exponent $\alpha$ and that there are no constraints on the reference
kernel radius $r_{\mathrm{kernel}}^0$ and reference number of particles per kernel radius $n_{\mathrm{ppk}}^0$. Even though it seems logical to require a minimum
number of particles per kernel radius so that enough neighbors are available to get a reasonable estimate of the gradient.
These parameters should be adjusted according to the expected accuracy of the results and/or the computer power available.
Determining the optimal parameter values for $\alpha$, $r_{\mathrm{kernel}}^0$ and $n_{\mathrm{ppk}}^0$, for example according to a user's needs in terms of accuracy
and computational efficiency, requires a specific and detailed investigation of the considered case. In the Sect. 5, we will explore
model convergence using the condition eq. 42 with different values of $\alpha$.

## 4 Numerical implementation / Solver

In this section, the numerical implementation and algorithm of the **com1DFA** module are described. The following sections
are organized following the workflow used in **com1DFA** code which is also illustrated in Fig. 2. First the release mass is
discretized into particles and the grid is initialized. As a result of the partial differential equations considered and the time
scheme used, stability issues might arise. Hence, artificial viscosity is added in order to ensure the stability of the solution. As



a next step, driving forces (including curvature effects) are accounted for. Friction forces are taken into account subsequently, in order to ensure proper starting and stopping behavior. Finally, a reprojection step is needed to ensure that particles lie on the topography and that particle velocities are tangent to the topography. For simplicity and because they are not considered in the verification tests in Sect. 5, entrainment and added resistance effects are not included in what follows. Additional information about entrainment or resistance forces is available in the theory section of the AvaFrame documentation.

### 4.1 Initialization

To start a simulation with **com1DFA**, input information about topography, material properties and initial conditions is required. Topography is described by a DEM (digital elevation model) which is a grid. It is defined by the (x,y) Cartesian location of the lower left cell center and a cell size, containing information about the elevation z. The material is defined by its density and some friction properties. The initial condition is given by release areas, polygons describing the initial material location, and the release thickness, in our case measured in the surface normal direction.

Then the material is discretized into particles. Surface normal vector fields are computed from the input DEM and the different grid vector fields are initialized. The details of the initialization process are given in the initialization section of the AvaFrame documentation.

### 4.2 Numerical stability

Because the lateral shear force term was removed when deriving the model equations (because of its relative smallness, Gray and Edwards, 2014), Eq. 22 is hyperbolic. Hyperbolic systems have the characteristic of carrying discontinuities or shocks which will cause numerical instabilities. They would fail to converge if for example an Euler forward in time scheme is used (LeVeque, 1990). Several methods exist to stabilize the numerical integration of an hyperbolic system of differential equations. All aim at adding some upwinding in the discretization scheme. Some methods tackle this problem by introducing some upwinding in the discretization of the derivatives (Harten et al., 1983; Harten and Hyman, 1983). Others introduce some artificial viscosity (as in Monaghan, 1992), which is also implemented in **com1DFA**. The following artificial viscosity force acting on particle $k$ is added to stabilize the momentum equation:

$$\begin{aligned}
\mathbf{F_k^{visc}} &= -\frac{1}{2}\rho_0 C_{Lat} A_k^{\text{Lat}} \|\mathbf{d\overline{u}}_k\|^2 \frac{\mathbf{d\overline{u}}_k}{\|\mathbf{d\overline{u}}_k\|} \\
&= -\frac{1}{2}\rho_0 C_{Lat} A_k^{\text{Lat}} \|\mathbf{d\overline{u}}_k\| \mathbf{d\overline{u}}_k,
\end{aligned} \tag{43}$$

where the velocity difference reads $\mathbf{d\overline{u}}_k = \overline{\mathbf{u}}_k - \widehat{\overline{\mathbf{u}}}_k$ ($\widehat{\overline{\mathbf{u}}}_k$ represents the averaged velocity of the neighbor particles and is practically the grid velocity interpolated at the particle position). $C_{Lat}$ is a coefficient that controls the viscous force. It would be the equivalent of $C_{Drag}$ in the case of the drag force. $C_{Lat}$ is a numerical parameter that depends on the grid size.

In this expression, let $\overline{\mathbf{u}}_k^n$ be the velocity at the beginning of the time step and $\overline{\mathbf{u}}_k^{n+1\blacktriangle}$ be the velocity after adding the numerical viscosity (Fig. 2). In the norm term $\|\mathbf{d\overline{u}}_k\|$ the particle and grid velocity at the beginning of the time step are used. This ensures no implicit relation on the norm term or on the average velocity $\widehat{\overline{\mathbf{u}}}_k$. On the contrary, an implicit formulation is





used in $\mathbf{d\overline{u}}_k$ because the new value of the velocity is used there. The artificial viscosity force now reads:

$$\mathbf{F_k^{visc}} = -\frac{1}{2}\rho_0 C_{Lat} A_k^{\mathrm{Lat}} \left\| \overline{\mathbf{u}}_k^n - \widehat{\overline{\mathbf{u}}}_k^n \right\| \left( \overline{\mathbf{u}}_k^{n+1\,\blacktriangle} - \widehat{\overline{\mathbf{u}}}_k^n \right) \tag{44}$$

Updating the velocity then gives:

$$\overline{\mathbf{u}}_k^{n+1\,\blacktriangle} = \frac{\overline{\mathbf{u}}_k^n - C_{vis}\widehat{\overline{\mathbf{u}}}_k^n}{1 + C_{vis}} \tag{45}$$

with

$$C_{vis} = \frac{1}{2}\rho_0 C_{Lat} A_k^{\mathrm{Lat}} \left\| \overline{\mathbf{u}}_k^n - \widehat{\overline{\mathbf{u}}}_k^n \right\| \frac{\Delta t}{m}. \tag{46}$$

This approach to stabilize the momentum equation (Eq. 30) is not optimal for different reasons. Firstly, it introduces a new coefficient $C_{vis}$ which is not a physical quantity and will require to be calibrated. Secondly, it artificially adds a force that should be described physically. So it would be more interesting to take the physical force into account in the first place.

Potential solutions could be taking the physical shear force into account, using for example the μ-I rheology (Gray and Edwards, 2014; Baker et al., 2016). Another option would be to replace the artificial viscosity with a purely numerical artifact aiming to stabilize the equations such as a SPH version of the Lax-Friedrich scheme as presented in Ata and Soulaïmani (2005).

### 4.3 Curvature acceleration term

The last term of the particular momentum equation (Eq. 30) as well as the effective gravity $g^{\mathrm{eff}}$ are the final terms to be discretized before the numerical integration. In both of these terms, the remaining unknown is the curvature acceleration term $\overline{\mathbf{u}}_k \cdot \frac{\mathrm{d}\mathbf{v}_{3,k}}{\mathrm{d}t}$. Using the forward Euler time discretization for the temporal derivative of the normal vector $\mathbf{v}_{3,k}$ gives:

$$\frac{\mathrm{d}\mathbf{v}_{3,k}}{\mathrm{d}t}\bigg|^n \approx \frac{\mathbf{v}_{3,k}^{n+1} - \mathbf{v}_{3,k}^n}{\Delta t} \tag{47}$$

$\mathbf{v}_{3,k}^n$ is a known quantity, the normal vector of the bottom surface at $\mathbf{x}_k^n$ wich is interpolated from the grid normal vector values at the position of the particle $k$ at time $t^n$. $\mathbf{v}_{3,k}^{n+1}$ is unknown since $\mathbf{x}_k^{n+1}$ is not known yet, hence we estimate $\mathbf{x}_k^{n+1}$ based the position $\mathbf{x}_k^n$ and the velocity at $t^n$:

$$\mathbf{x}_k^{n+1} = \mathbf{x}_k^n + \Delta t\, \overline{\mathbf{u}}_k^{n+1\,\blacktriangle} \tag{48}$$

This position at $t^{n+1}$ is projected onto the topography and $\mathbf{v}_{3,k}^{n+1}$ can be interpolated from the grid normal vector values.

Note that the curvature acceleration term is needed to compute the bottom pressure (Eq. 24), which is used for the bottom friction computation and for the pressure gradient computation. The curvature acceleration term can lead to a negative value, which means detachment of the particles from the bottom surface. In **com1DFA**, surface detachment is not allowed and if pressure becomes negative, it is set back to zero forcing the material to remain in contact with the topography.



### 4.4 Account for driving forces

Adding the driving forces is done after adding the artificial viscosity as described on Fig. 2. The velocity is updated as follows ($\overline{\mathbf{u}}_k^{n+1\,\star}$ is the velocity after taking the driving force into account):

$$\overline{\mathbf{u}}_k^{n+1\,\star} = \overline{\mathbf{u}}_k^{n+1\,\blacktriangle} + \frac{\Delta t}{m_k}\mathbf{F}_k^{\text{drive}} = \overline{\mathbf{u}}_k^{n+1\,\blacktriangle} + \frac{\Delta t}{m_k}\left(-m_k\, g_k^{\text{eff}}\,\boldsymbol{\nabla}_s h + m_k\mathbf{g}_s - m_k\left(\overline{\mathbf{u}}_k^{n+1\,\blacktriangle}\cdot\left.\frac{\mathrm{d}\mathbf{v}_{3,k}}{\mathrm{d}t}\right|^n\right)\mathbf{v}_{3,k}^n\right) \tag{49}$$

### 4.5 Account for friction forces

The friction force related to the bottom shear force needs to be taken into account for in the momentum equation and the velocity needs to be updated accordingly. Friction force acts against motion, hence it only affects the magnitude of the velocity and can not be a driving force (Mangeney-Castelnau et al., 2003). Moreover, the friction force magnitude depends on the particle state, i.e. if it is flowing or at rest. If the velocity of the particle $k$ reads $\overline{\mathbf{u}}_k^{n+1\,\star}$ after adding the driving forces, adding the friction force leads, depending on the sign of $\frac{m_k\left\|\overline{\mathbf{u}}_k^{n+1\,\star}\right\|}{\Delta t} - \left\|\mathbf{F}_k^{\text{fric}}\right\|_{max}$ (where $\left\|\mathbf{F}_k^{\text{fric}}\right\|_{max}$ depends on the chosen friction law introduced in Sect. 3.4), to:

- $\left\|\mathbf{F}^{\text{fric}}\right\| = \left\|\mathbf{F}_k^{\text{fric}}\right\|_{\text{max}}$ and $\overline{\mathbf{u}}_k^{n+1} = \overline{\mathbf{u}}_k^{n+1\,\star}\left(1 - \frac{\Delta t}{m_k}\frac{\left\|\mathbf{F}_k^{\text{fric}}\right\|_{\text{max}}}{\left\|\overline{\mathbf{u}}_k^{n+1\,\star}\right\|}\right)$, if $\frac{m_k\left\|\overline{\mathbf{u}}_k^{n+1\,\star}\right\|}{\Delta t} > \left\|\mathbf{F}_k^{\text{fric}}\right\|_{\text{max}}$

- $\left\|\mathbf{F}_k^{\text{fric}}\right\| \leq \left\|\mathbf{F}_k^{\text{fric}}\right\|_{\text{max}}$ and the particle stops moving $\overline{\mathbf{u}}_k^{n+1} = 0$ before the end of the time step, if $\frac{m_k\left\|\overline{\mathbf{u}}_k^{n+1\,\star}\right\|}{\Delta t} \leq \left\|\mathbf{F}_k^{\text{fric}}\right\|_{\text{max}}$

This method prevents the friction force to become a driving force and nonphysically change the direction of the velocity. This would lead to oscillations of the particles instead of stopping. Adding the friction force following this approach (Mangeney-Castelnau et al., 2003) allows the particles to start and stop flowing properly.

### 4.6 Reprojection

The last term in Eq. 30 (accounting for the curvature effects) adds a non tangential component allowing the new velocity to lie in a different plane than the one from the previous time step. This enables the particles to follow the topography. But because the curvature term was only based on an estimation (see Sect. 4.3), it can happen that the new particle position is not necessarily on the topography and the new velocity does not necessarily lie in the tangent plane at this new position. Furthermore, in case of a strong convex curvature and high velocities, the particles can theoretically be in a free fall state (detachment) as mentioned in Sect. 2.5.1. **com1DFA** does not allow detachment of the particles and the particles are forced to stay on the topography. This consists in a limitation of the model/method which will lead to nonphysical behaviors in special cases (material flowing over a cliff). In both of the previously mentioned cases, the particles positions are projected back onto the topography and the velocity direction is corrected to be tangential to the topography. The position reprojection is done using an iterative method that attempts to conserve the distance traveled by each particle between $t^n$ and $t^{n+1}$. The velocity reprojection changes the direction of the velocity but its magnitude is conserved.



**Figure 2.** Dense Flow Avalanche solver (**com1DFA**) workflow.



## 5 Model Verification

In this section, the numerical implementation of the mathematical model is tested. We present different tests where, for specific
400 conditions, an (semi-) analytical solution exists. The tests described here are implemented in the **ana1Tests** module from
AvaFrame. In the first set of tests, the flow variable tests, we compare the temporal and spatial evolution of the flow thickness
($h$) and the thickness integrated flow momentum flux ($h\overline{\mathbf{u}}$) of the **com1DFA** simulation results to an (semi-) analytical solution.
With these tests, we aim at verifying the numerical discretization and implementation of the solver as well as checking the
validity of the convergence criterion described in Sect. 3.6.

405 In the second test, the energy line test, we investigate global variables such as mass averaged position and kinetic energy that
are derived from the DFA simulations. This test is based on energy conservation considerations for simplified topographies.
This allows to verify the accuracy of the DFA simulations in terms of mass averaged runout. All the tests presented and used
in what follows are implemented and available in AvaFrame (both data and helper functions). All results and figures can be
reproduced using the code available on the AvaFrame github repository.

410 ### 5.1 Flow variable tests

Before performing the similarity solution and dam break tests, it is necessary to describe the quantities that are compared and
the measures that are used to assess the convergence, accuracy or precision of the numerical model. Both the flow thickness
($h$) and thickness integrated flow momentum flux ($h\overline{\mathbf{u}}$) are used to compare the analytical solution to the simulation results.
Two different deviation measures are used to quantify the deviation between a reference solution and the simulation result on a
415 domain (one or two dimensional). The first one is based on the $\mathcal{L}_{\max}$ norm (uniform norm), the second on the Euclidean norm
($\mathcal{L}_2$ norm). Let $f_{\text{num}}$ be the numerical solution and $f_{\text{ref}}$ the reference solution defined on a domain $\Omega$. The local deviation is
defined by $\mathcal{E}(x) = f_{\text{num}}(x) - f_{\text{ref}}(x)$ and the global deviation by:

– The uniform norm ($\mathcal{L}_{\max}$) measures the largest absolute value of the deviation $\mathcal{E}$ on $\Omega$:

$$\mathcal{L}_{\max}(\mathcal{E}) = \sup_{x \in \Omega}(|\mathcal{E}(x)|)$$

This norm is applied to one or two dimensional results. It can also be normalized by dividing the uniform norm of the
deviation by the uniform norm of the reference. In this case we refer to the relative deviation:

$$\mathcal{RL}_{\max} = \frac{\mathcal{L}_{\max}(\mathcal{E})}{\mathcal{L}_{\max}(f_{\text{ref}})}$$

– The Euclidean norm ($\mathcal{L}_2$ norm) gives an overall measure of the deviations

$$\mathcal{L}_2(\mathcal{E}) = \int_{x \in \Omega} ||\mathcal{E}(x)||^2 \, dx$$

It is useful to normalize the norm of the deviation either by dividing with the norm of the reference solution:

$$\mathcal{RL}_2 = \frac{\mathcal{L}_2(\mathcal{E})}{\mathcal{L}_2(f_{\text{ref}})}$$





or by the measure of the interval ($\mathcal{L}_2(1) = \int_{x \in \Omega} dx$):

$$\mathcal{RSL}_2 = \sqrt{\frac{\mathcal{L}_2(\mathcal{E})}{\mathcal{L}_2(1)}}$$

The first normalization approach will give a relative deviation in % whereas the second will give an average deviation of $f$ on $\Omega$.

### 5.1.1 Similarity solution test

The similarity solution is one of the few cases where a semi-analytic solution is available for solving the thickness integrated equations. This makes it very useful for validating the implementation of dense flow avalanche numerical methods (here **com1DFA**). In this problem, we consider an avalanche governed by a dry friction law (Coulomb friction) flowing down an inclined plane. Release mass is initially distributed in an ellipse (diameters of length $L_x$ and $L_y$) with a parabolic thickness shape ($H$ in the middle). This mass is released at $t = 0$ and flows down the inclined plane, as illustrated in Fig. 3a for the initial time step and at some later time $t$. This semi-analytic solution can be derived under very strict conditions: First a relative smallness of the ratio $\epsilon = H/L \ll 1$ of flow thickness over spatial extent in flow and cross flow direction. Second an inequality between the slope angle and friction angle. Third by making one major assumption on the shape of the solution: symmetry/anti-symmetry of the solution around the x and y axis. The full description of the conditions and assumptions as well as the derivation of the solution is presented in detail in Hutter et al. (1993). The term semi-analytic is used here because the method enables to transform the PDE (partial differential equation) of the problem into an ODE (ordinary differential equation) using a similarity analysis method. However, solving the ODE still requires a numerical integration but this one is more accurate (when conducted properly, here using an explicit Runge-Kutta method of fourth order) and requires less computational power than solving the PDE.

This test is implemented in the **ana1Tests** module of AvaFrame which offers functions to compute the semi-analytic solution, to compare it to the output from the DFA computational module and to visualize the results.

### 5.1.2 Dam break test

The dam break test is the second test for which an analytical solution of the thickness integrated equations is known. In this test, we also consider an avalanche governed by a dry friction law (Coulomb material), released from rest on an inclined plane (see Fig. 3b). In the case of a thickness integrated model as derived by Savage and Hutter (e.g. in Hutter et al., 1993), an analytical solution exists under the assumption of shallowness of the flow. Furthermore the friction angle has to be smaller than the slope of the inclined plane. This test, in contrast to the similarity solution test focuses on the very early stages of the flow and not on the evolution over time and lateral spreading. The derivation of the dam break solution is described in Faccanoni and Mangeney (2013) and corresponds to a Riemann problem. It has the following initial conditions:



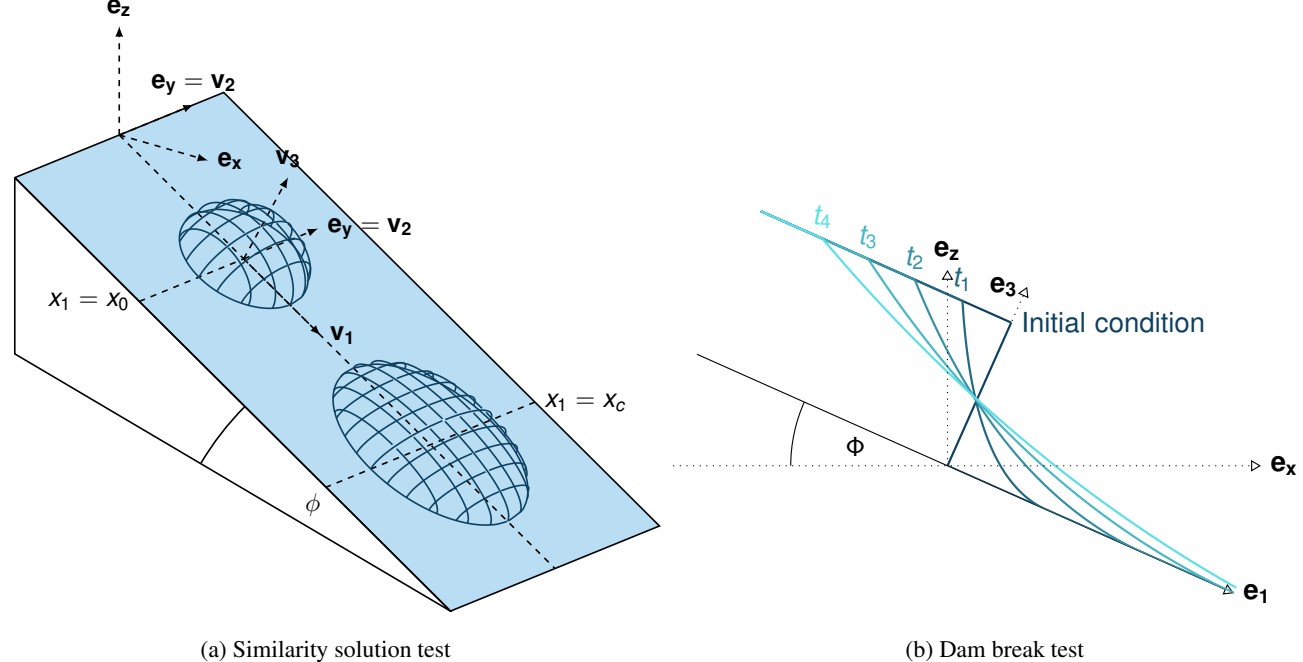

(a) Similarity solution test            (b) Dam break test

**Figure 3.** Flow variable test setups: (a) 3D view of the material on the inclined plane at $t = 0$ (center located in $x_1 = x_0$) and at a general time $t$ (center located in $x_1 = x_c$). The footprint of the material on the inclined plain at the initial time is an ellipse of diameters $L_x$ and $L_y$, the thickness follows a parabolic distribution with the maximum thickness of $H$ in the middle of the ellipse and 0 on the edge, (b) 2D view of the material on the inclined plane (profile in flow direction) at the $t = 0$ and for different $t > 0$.

$$445 \quad (h, \mathbf{u})(x, t = 0) = \begin{cases} (h_0, \mathbf{0}), & \text{if } x \leq 0 \\ (0\ , \mathbf{0}), & \text{if } x > 0 \end{cases} \tag{50}$$

The dam break assumes an invariance in the $y$ direction which is achieved using a wide enough domain in the $y$ direction so that lateral effects can be neglected.

### 5.1.3 Results

DFA simulations are computed using the **com1DFA** module in AvaFrame varying the different numerical parameters listed in
450 Tab. 1. The range of $\alpha$ values (called "aPPK" in the code) is determined by the convergence criterion (Eq. 42). The SPH kernel radius $r_{\text{kernel}}$ (called "sphKernelRadius" in the code) is varied around 5m which is the raster cell size value currently used for operational hazard mapping in Austria. The range of $n_{ppk0}$ (called "nPPK0" in the code) and $C_{\text{time}}$ (called "cMax" in the code) used are listed in the Tab. 1. A large $n_{ppk0}$ and small $C_{\text{time}}$ lead to very long computation time which makes it unrealistic and impractical to use. Instructions on how to reproduce the results presented below are provided in the supplementary material.



| test | sphKernelRadius | nPPK0 | aPPK | cMax |
|---|---|---|---|---|
| similarity solution | $\{\mathbf{10}, \mathbf{8}, \mathbf{6}, \mathbf{5}, \mathbf{4}, \mathbf{3}\}$ | $\{\mathbf{15}, 20, 30, 40\}$ | $\{\mathbf{0}, -\mathbf{0.5}, -\mathbf{1}, -1.5, -\mathbf{2}, -2.5, -\mathbf{3}\}$ | $\{0.04, \mathbf{0.02}, 0.01, 0.005\}$ |
| dam break | $\{\mathbf{10}, \mathbf{8}, \mathbf{6}, \mathbf{5}, \mathbf{4}, \mathbf{3}\}$ | $\{\mathbf{15}, 20, 30\}$ | $\{\mathbf{0}, -\mathbf{0.5}, -\mathbf{1}, -1.5, -\mathbf{2}, -2.5, -\mathbf{3}\}$ | $\{0.04, \mathbf{0.02}, 0.01, 0.005\}$ |

**Table 1.** Parameter variation used to study convergence of the DFA simulation solution for both similarity solution and dam break test (in bold the parameters used for the figures presented in this manuscript).

For both of the tests, the numerical schemes to apply friction and the method used to compute the SPH gradient are crucial to capture a proper starting and behavior of the flow. Some intermediate developments showed that adding the friction with different methods as the one presented here and computing the SPH gradient without taking the slope inclination into account leads to unsatisfying results. This is why the friction force is added as described in Sect. 4.5 and the SPH force is computed as described in Appendix B2. In what follows, artificial viscosity is added (in the following results the $C_{\text{Lat}} = 10$) in the similarity solution test, which stabilizes the solution without degrading the match with the semi-analytic solution. For the dam break test, adding artificial viscosity has a negative impact on the solution and the following results were produced with no artificial viscosity ($C_{\text{Lat}} = 0$).

Fig. 4 shows an example where a DFA simulation is compared to the semi-analytical solution of the similarity solution test case. The chosen parameters for this example are $r_{\text{kernel}} = 3$m, $\alpha = -3$, $n_{ppk}^0 = 15$ and $C_{\text{time}} = 0.02$s and correspond to the most accurate of the simulations presented here. The reader can find the results of the similarity solution test with the standard parameters in the supplementary material. The top panels show the flow thickness and momentum profiles in and across flow direction after 20 seconds of flow. The bottom right panel shows the evolution of the $\mathcal{RL}_2$ and $\mathcal{RL}_{\text{max}}$ deviation with time. The deviation at the initial time step ($t = 0s$) is rather high (this is related to the random process to initialize the particles in the simulation) and then quickly decreases after a few seconds of simulation due to the reorganization of the particles. The deviation then increases again as the numerical inaccuracies accumulate. When varying the numerical parameters in the DFA simulations (according to Eq. 41), the computed $\mathcal{L}_2$ deviations between DFA results and the semi-analytical solution decrease (see Fig. 5). In Fig. 6, the comparison between a DFA simulation and the analytical solution of the dam break test is shown for flow thickness, flow velocity and momentum at t=15s (upper panel). The lower left panel shows a top view of the flow colored by flow thickness. This panel also shows the domain on which the deviation between analytical- and numerical solution is computed. The lower right panel shows the relative deviations $\mathcal{RL}_2$ and $\mathcal{RL}_{\text{max}}$ on flow thickness and momentum. The same behavior as for the similarity solution test is observed regarding the time evolution of the deviation. Computation was done with $r_{\text{kernel}} = 3$m, $\alpha = -3$, $n_{ppk}^0 = 15$ and $C_{\text{time}} = 0.02$s. The reader can find the results of the similarity solution test with the standard parameters in the supplementary material.

Results from both the similarity solution tests and the dam break test validate the convergence criterion from Moussa and Vila (2000). Indeed, with a $\alpha$ exponent smaller than $-1$, decreasing the SPH kernel radius and varying the other parameters according to Eq. 41 leads to a decrease in the deviation. Whereas for larger exponents, $\alpha = -0.5$ for both tests or $\alpha = 0$ or the dam break test (Fig. 7), the decrease is weak or even an increase of the deviation is observed. Moreover, it is observed




(not shown on the figure) that decreasing the time step (decreasing the $C_{\text{max}}$ parameter) with all other parameters fixed leads to a decreasing deviation. Finally, for these two specific cases, DFA simulation results converge towards the semi-analytical or
analytical solution.

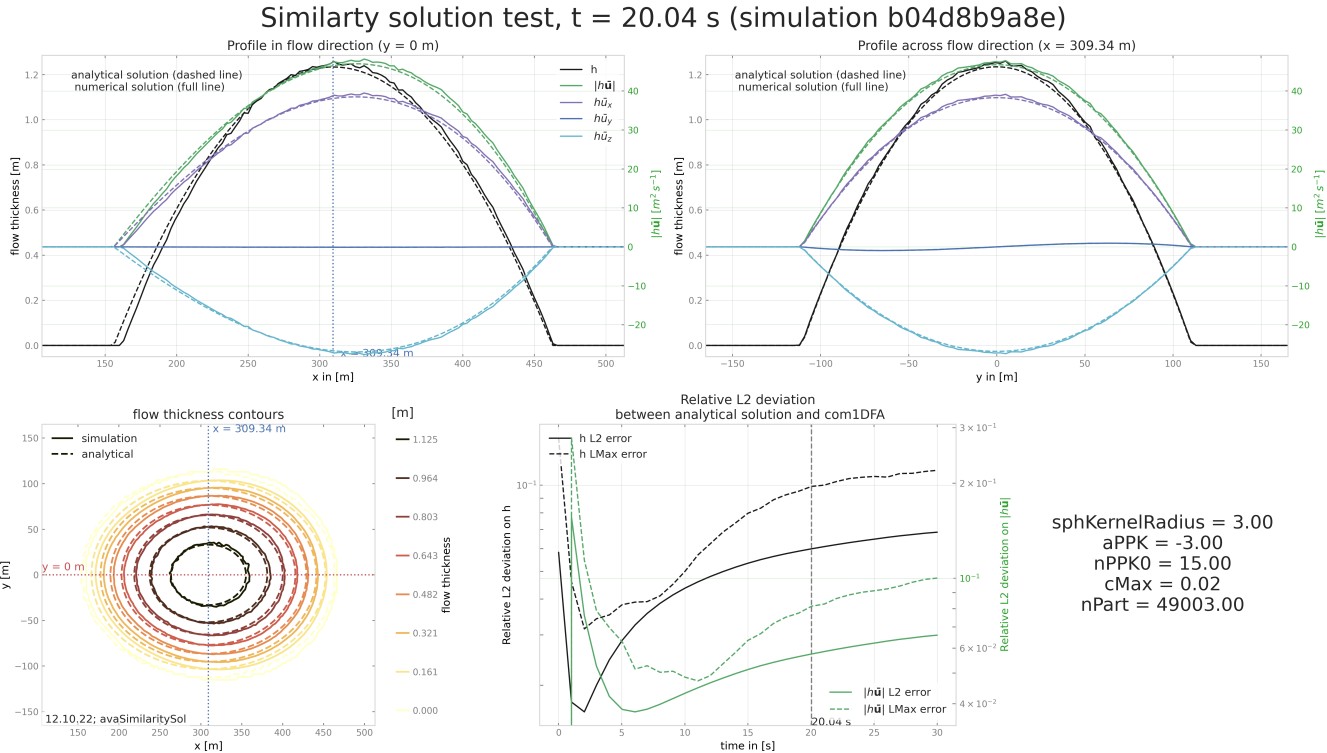

**Figure 4.** Comparison of the analytic (dashed lines) and numerical (solid lines) solution for the similarity solution test case at $t = 20.04s$. The upper Left panel shows the profile in the flow direction (along the x axis) whereas lower right panel shows the profiles across flow direction (along the y axis). The lower left panel provides a top view of the flow thickness (flow thickness contour lines). The lower right panel shows the time evolution of the deviation (both $\mathcal{RL}_{\text{max}}$ and $\mathcal{RL}_2$) on flow thickness $h$ and momentum $\|h\overline{\mathbf{u}}\|$ between analytic and numerical solution ($r_{\text{kernel}} = 3\text{m}$, $\alpha = -3$, $n_{ppk}^0 = 15$ and $C_{\text{time}} = 0.02\text{s}$).

## 5.2 Energy and runout testing

The Energy line test compares the results of the **com1DFA** simulation to a geometrical solution derived from the total energy of the system. Solely considering Coulomb friction, this solution is motivated by the first principle of energy conservation along a simplified topography. In this case, the friction force only depends on the slope angle. The analytical runout is the intersection
of the path profile with the geometrical line ($\alpha$ line) defined by the friction angle $\alpha$. From the geometrical line it is also possible to extract information about the flow mass averaged velocity at any time or position along the path profile.



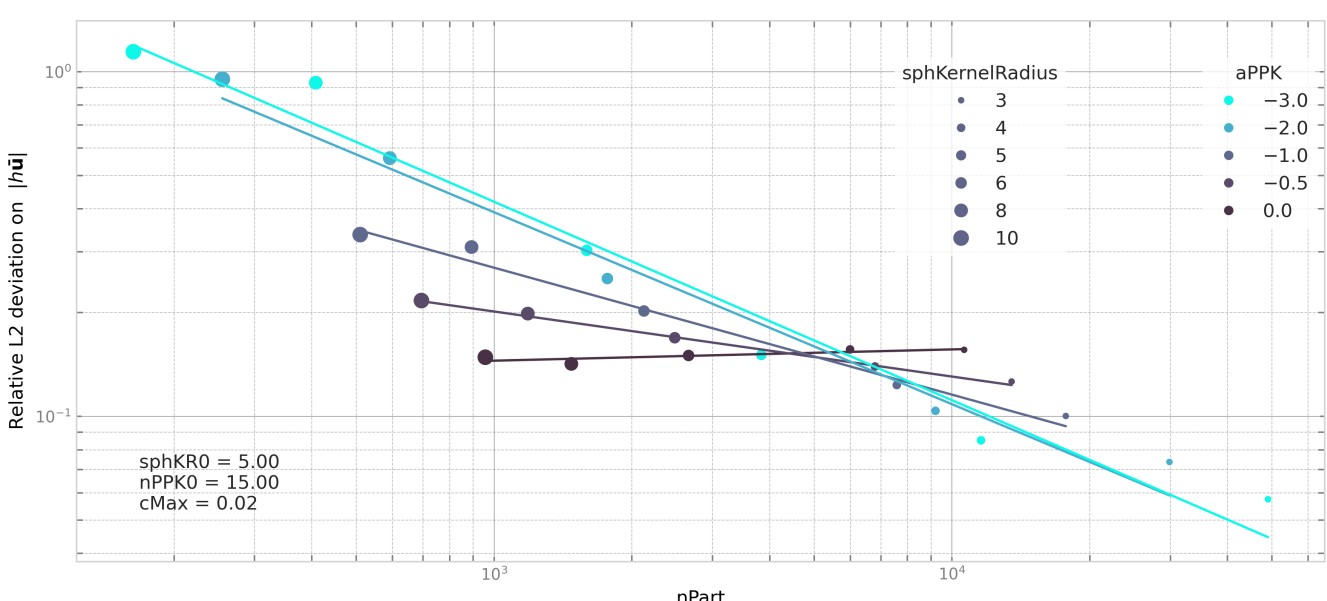

**Figure 5.** $\mathcal{R}\mathcal{L}_2$ deviation on flow momentum in the similarity solution test case for different $\alpha$ exponents and SPH kernel radius $r_{ppk}$. Other parameters are kept fixed (reference kernel radius $r_{ppk}^0$, time constant $C_{\text{time}}$ and reference number of particles per kernel radius $n_{ppk}^0$). The points corresponding to simulations with $r_{ppk} = 5\,\text{m}$ are all overlapping since they are all identical. The colored lines are added to give an idea of the convergence speed trend associated to each $\alpha$ scenario. One can observe that decrease in deviation is stronger for lower $\alpha$ exponents and that no or little decrease is observed for $\alpha = 0$ or $\alpha = -0.5$.

### 5.2.1  Theory

The conservation of energy for a material point (block model) flowing down slope assuming only coulomb friction is at work, reads between two points O and B:

$$E_B^{tot} - E_O^{tot} = E_B^{kin} + E_B^{pot} - (E_O^{kin} + E_O^{pot}) = \int_O^B \delta E_{fric}$$

$$= \frac{1}{2}mv_B^2 + mgz_B - \frac{1}{2}mv_O^2 - mgz_O \tag{51}$$

$$= \int_O^B \mathbf{F}_{fric} \cdot \mathbf{dl} = -\int_O^B \mu||\mathbf{N}||\frac{\mathbf{dl}}{dl} \cdot \mathbf{dl} = -\int_O^B \mu mg(\mathbf{e_z} \cdot \mathbf{n})dl$$

where $\delta E_{fric}$ is the energy dissipation due to friction, $\mathbf{N}$ represents the normal (to the bottom surface) component of the gravity force, $\mathbf{n}$ the normal vector to the slope surface and $\mathbf{dl}$ is the elementary vector on the path profile traveled by the material point between O and B. The vertical component of the normal vector reads $\mathbf{e_z} \cdot \mathbf{n} = cos(\theta)$, where $\theta$ is the slope angle. $m$ represents



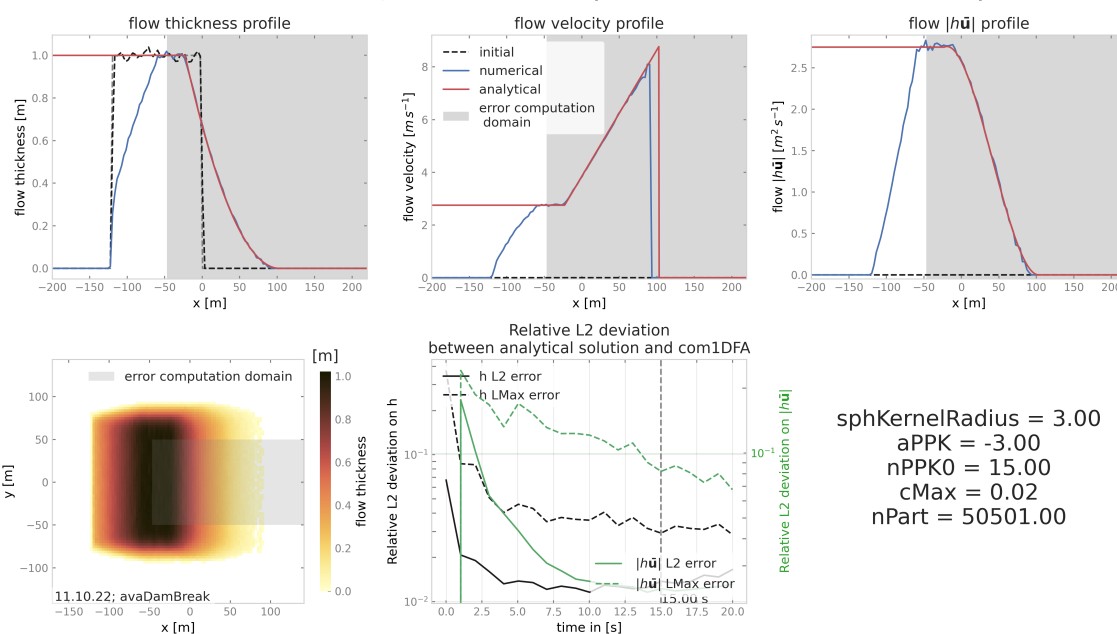

**Figure 6.** Comparison of the analytic and numerical solution for the dam break test caseat $t = 15.0s$. Top panel shows flow thickness, velocity and momentum profiles in flow direction. The lower left panel provides a top view of the flow thickness for the dam break. The gray shaded rectangle represents the domain on which the deviations are computed. The lower right panel shows the time evolution of the deviation (both $\mathcal{RL}_{\max}$ and $\mathcal{RL}_2$) on flow thickness $h$ and momentum $\|h\overline{\mathbf{u}}\|$ between analytic and numerical solution ($r_{\mathrm{kernel}} = 3$m, $\alpha = -3$, $n_{ppk}^0 = 15$ and $C_{\mathrm{time}} = 0.02$s).

the mass of the material point, $g$ the gravity, $\mu = \tan\alpha$ the friction coefficient and friction angle, $z$ the elevation and $v$ the

velocity of the material point. Note that in the 2D case, $dl = \frac{ds}{cos(\theta)}$, which means that the material point is flowing in the steepest slope direction (**ds** is the horizontal component of **dl**). Now considering O as the origin position ($s_O = 0$ and $v_O = 0$) leads to the following simplification:

$$
\begin{aligned}
E_B^{tot} - E_O^{tot} &= \frac{1}{2}mv_B^2 + mgz_B - \frac{1}{2}m\cancelto{0}{v_O^2} - mg\cancelto{z_0}{z_O} \\
&= \int_O^B \delta E_{fric} = -\int_{s'=s_O}^{s'=s_B} \mu mg\, ds' = -\mu mg(s_B - \cancelto{0}{s_O})
\end{aligned}
\tag{52}
$$

Speaking in terms of altitude, the energy conservation equation can be rewritten (the subscript B is dropped):

$$z_O = z + \frac{v^2}{2g} + \tan\alpha \tag{53}$$



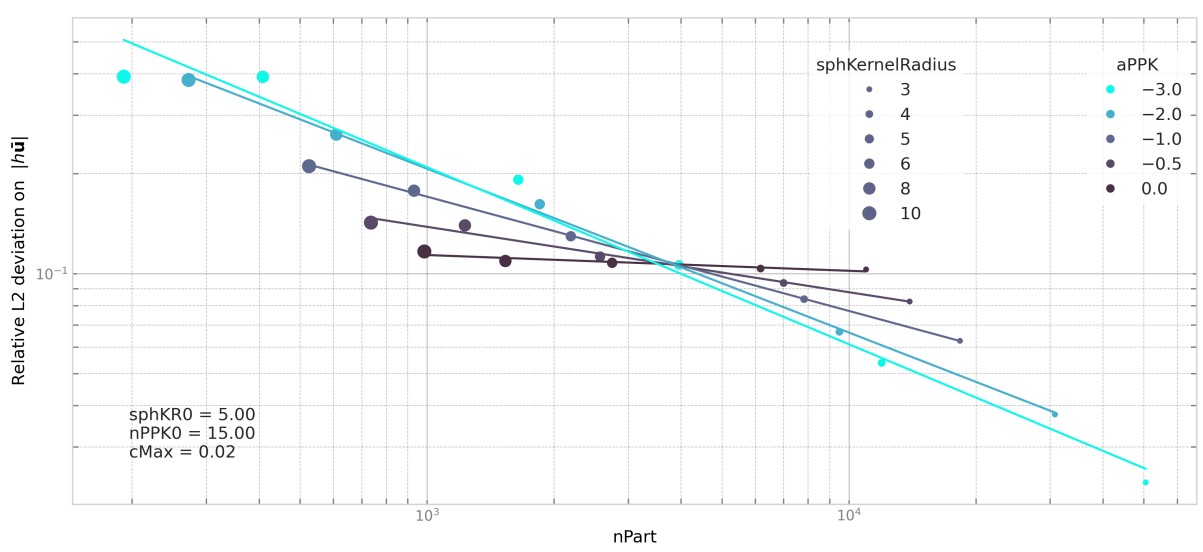

**Figure 7.** $\mathcal{R}\mathcal{L}_2$ deviation on flow momentum in the dam break case for different $\alpha$ exponents and SPH kernel radius $r_{ppk}$. Other parameters are kept fixed (reference kernel radius $r_{ppk}^0$, time constant $C_{\text{time}}$ and reference number of particles per kernel radius $n_{ppk}^0$). The points corresponding to simulations with $r_{ppk} = 5$m are all overlapping since they are all identical. The colored lines are added to give an idea of the convergence speed trend associated to each $\alpha$ scenario. One can observe that decrease in deviation is stronger for lower $\alpha$ exponents and that no or little decrease is observed for $\alpha = 0$ or $\alpha = -0.5$.

Considering a system of material points flowing down a slope with Coulomb friction, we can sum the previous equation (Eq. 53) of each material point after weighting it by its mass. This leads to the mass averaged energy conservation equation:

$$\overline{z}_O = \overline{z} + \frac{\overline{v^2}}{2g} + \overline{s}\tan\alpha \tag{54}$$

where the mass averaged value $\overline{a}$ of a quantity $a$ is (k indicates the points indices):

$$\overline{a} = \frac{\sum_k m_k a_k}{\sum_k m_k} \tag{55}$$

This way, we can define the center of mass path $(\overline{x}, \overline{y})$ and the center of mass path profile $(\overline{s}, \overline{z})$. The mass averaged quantities also follow the same energy conservation law when expressed in terms of altitude. This result is illustrated in Fig. 8 and applies to both the material point equation (Eq. 53) and the mass averaged energy conservation equation (Eq. 54). The light blue line in Fig. 8 is obtained by evaluating the mass averaged energy conservation (Eq. 54) at the final time ($t_{\text{end}}$) and position (($\overline{s}_{\text{end}}$, $\overline{z}_{\text{end}}$)), where $v^2 = 0$. This leads to the $\alpha$ line (also called energy line) equation:

$$\overline{z} = \overline{z}_O - \overline{s}\tan\alpha \tag{56}$$

In this test, we use the $\alpha$ line to evaluate the DFA simulation. Computing the mass averaged path profile for the particles (each particle corresponding to a material point) in the simulation and comparing it to the $\alpha$ line allows us to compute four error





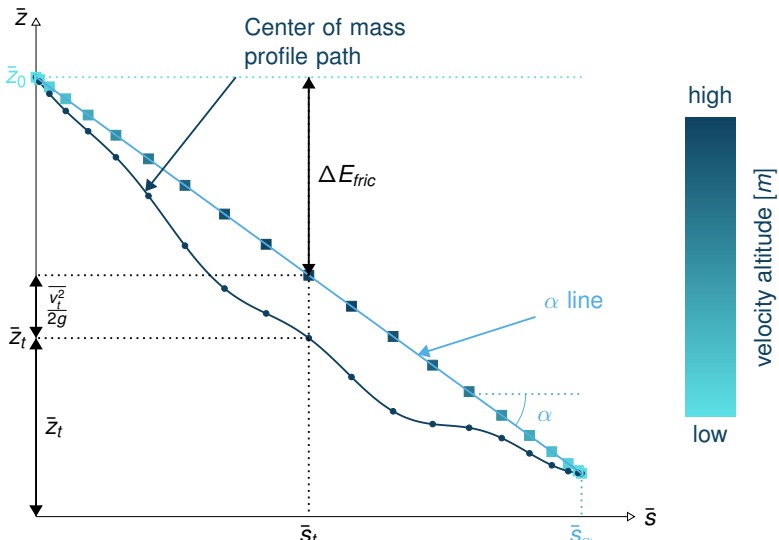

**Figure 8.** Center of mass profile (dark blue line with the dots) with on top, the $\alpha$ line (light blue) and the velocity altitude points (colored squares).

indicators. Fig. 9 illustrates the concept. The first three are related to the analytical runout point defined by the intersection
between the $\alpha$ line and the mass averaged path profile. The last one is related to the velocity:

- The horizontal distance between the analytical runout point and the end of the path profile defines the $\epsilon_s = \overline{s}_\gamma - \overline{s}_\alpha$ error in meters.

- The vertical distance between the analytical runout point and the end of the path profile defines the $\epsilon_z = \overline{z}_\gamma - \overline{z}_\alpha$ error in meters.

- The runout angle difference between the $\alpha$ line angle and the DFA simulation runout line defines the $\epsilon_\alpha = \gamma - \alpha$ runout angle error.

- The Root Mean Square Error (RMSE) between the $\alpha$ line and the DFA simulation energy points defines an error on the velocity altitude $\frac{\overline{v^2}}{2g}$.

**5.2.2  Limitations and applicability**

It is essential to state where the assumptions of this test hold. One of the important hypotheses when developing the energy solution is the material point flowing in the steepest slope direction, i.e. where $dl = \frac{ds}{cos(\theta)}$. If this hypothesis fails, as illustrated in Fig. 10, it is not possible to derive the analytic energy solution. In the 3D case, the distance vector $\mathbf{dl}$ traveled by the particles



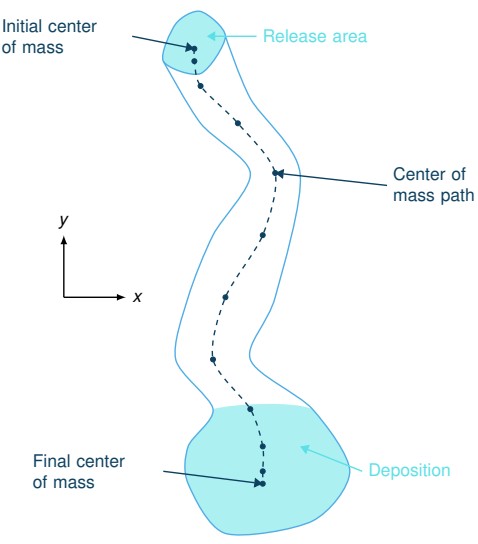

(a) View of the avalanche simulation and extracted path

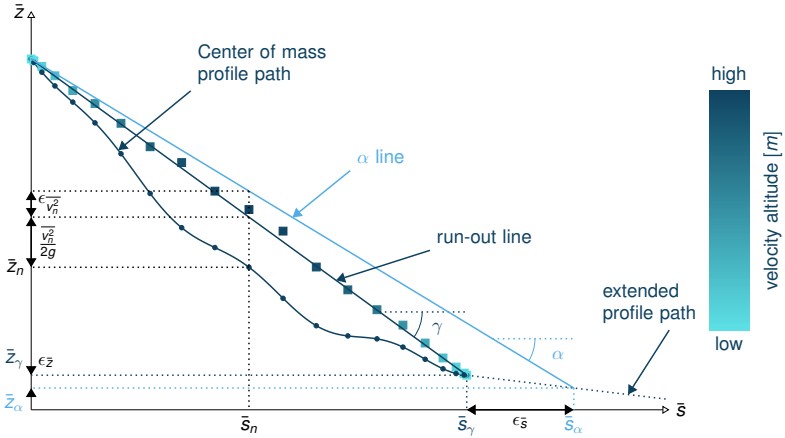

(b) Simulation path profile (dark blue curve and dots) with the runout line (dark blue line and velocity altitude squares), $\alpha$ line and runout error indicators ($\epsilon_s$ and $\epsilon_z$)

**Figure 9.** Mass averaged path and profile

reads $dl = \frac{\mathrm{d}s}{cos(\gamma)}$, where $\gamma$ is the angle between the $\mathbf{dl}$ vector and the horizontal plane which can differ from the slope angle $\theta$ ($\gamma \le \theta$). Here the energy solution is not the solution to the problem and hence can not be used as reference. In this case,
it would not be possible to distinguish what deviation is caused by the numerical error or because of the hypothesis being violated. However, even if this hypothesis is not satisfied, by e.g. not traveling in steepest slope or using another friction type,

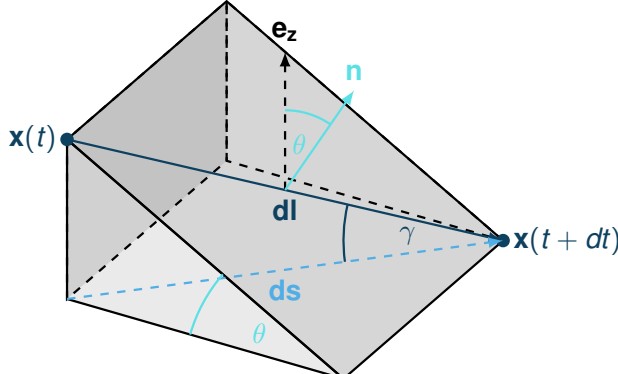

**Figure 10.** Example of trajectory where the steepest descent path hypothesis fails. The mass point is traveling from $\mathbf{x}(t)$ to $\mathbf{x}(t+dt)$. The slope angle $\theta$ and travel angle $\gamma$ are also illustrated. Here $(\mathbf{e_z} \cdot \mathbf{n})dl = cos\theta \frac{\mathrm{d}s}{cos\gamma} \neq ds$.

the energy line solution is still useful. The $\alpha$ line can be used to study the effect of terms such as curvature acceleration,



artificial viscosity or pressure gradients. For example, the curvature acceleration modifies the friction term, depending on topography curvature and particle velocity. This leads to a mismatch between the energy solution and the DFA simulation.

Artificial viscosity leads to viscous dissipation leading to shorter runouts than what the energy solution predicts. Finally effects of the pressure force can be studied and question like the two following ones can be answered: Does including the pressure gradient shorten or increase the runout? How do different methods to compute the pressure force affect the simulations? Fig. 12 shows an example with pressure forces and in this case it seems adding the pressure force does not influence the simulation runout. This can be explained by the fact that pressure forces do not dissipate any energy and hence should not affect the energy

balance. However pressure forces lead to particle trajectories that do not necessarily follow the steepest direction which means that the fundamental hypothesis illustrated in Fig. 10 is not satisfied. Figure 13 shows the effect of curvature. The topography considered here is an inclined plane that smoothly transitions into a horizontal plane, so curvature only occurs in the transition part. The energy line test for this case shows that there is added dissipation only in the transition part, seen by squares not following the $\alpha$ line. Once all particles have reached the horizontal plane, the squares follow the $\alpha$ line again (with a shift in s

coordinate).

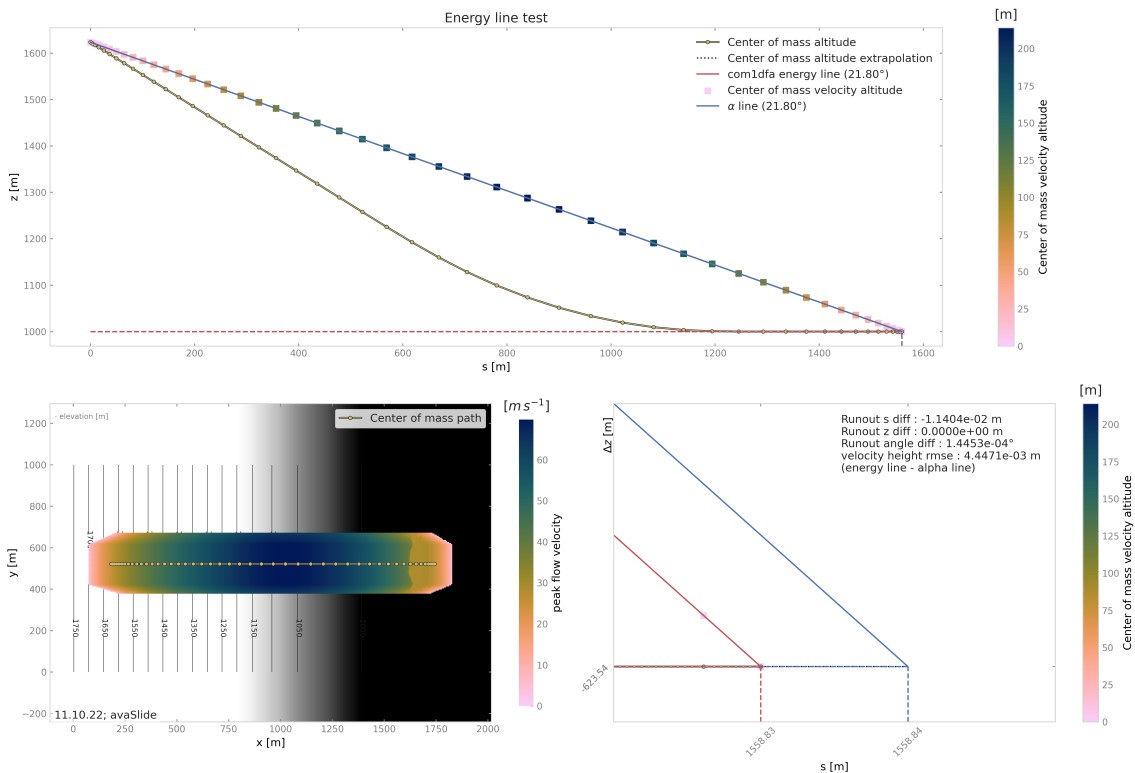

**Figure 11.** Energy line test for an inclined plane smoothly transitioning to a horizontal plane with particles following Coulomb friction and not subject to pressure forces. All hypothesis are satisfied and the geometrical solution can be used as reference to compute the numerical error that was commit (here less than $10^{-2}$m and $10^{-4\circ}$).



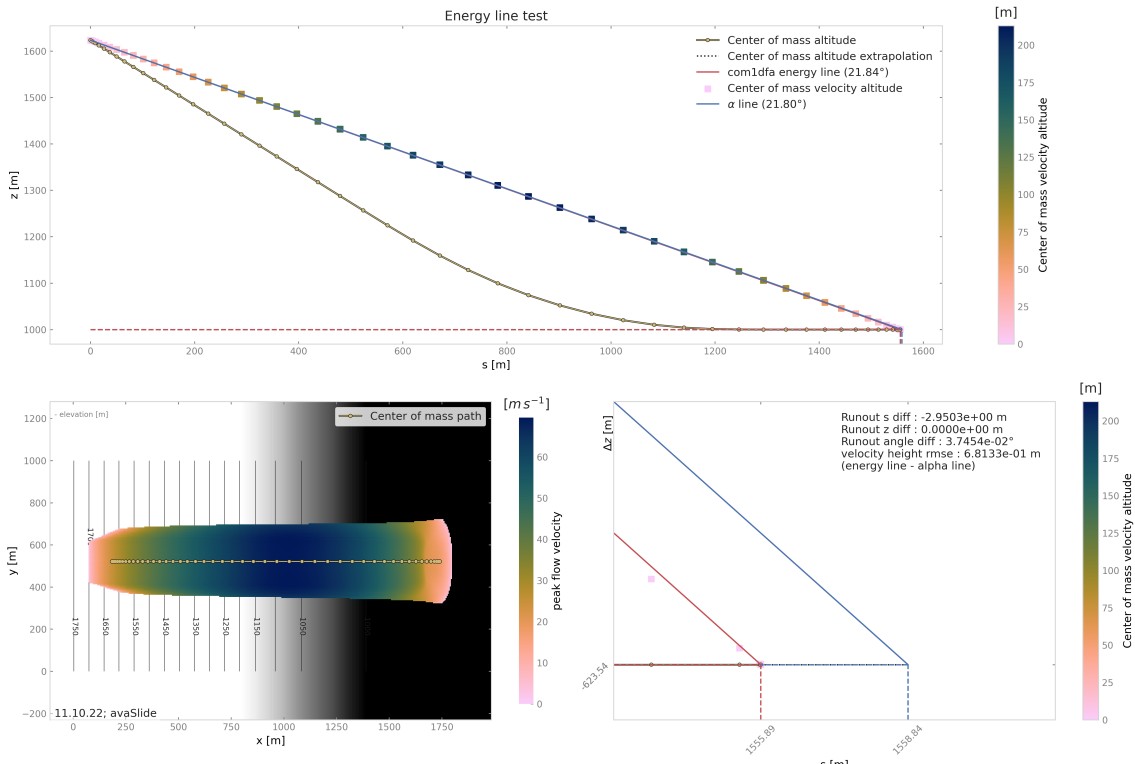

**Figure 12.** Energy line test for an inclined plane smoothly transitioning to a horizontal plane with particles following Coulomb friction and subject to pressure forces. In this case, the particles are flowing in the steepest direction (some go also in the y direction). This means that the geometrical solution can not be used as reference to compute the numerical error (or at least that the deviation between the geometric solution and the numerical solution is a combination of numerical errors and the failing hypothesis.

### 5.2.3 Grid orientation effect

The energy line test previously described is also used to test if the numerical method implemented in **com1DFA** performs independently of the grid orientation. Indeed we saw in Sect. 3.1 that **com1DFA** uses a regular grid to update some variables such as flow thickness or flow velocity. In order to quantify the effect of the grid orientation on the simulation results, we

perform tests where grid orientation is changed while keeping the grid cell size and topography the same. Here we show two examples where we consider a parabolic slope, i.e. the topography varies only in one direction and a bowl shape, i.e. the topography with a rotational symmetry about its center. The main axis of the flow is not always aligned with the grid and we provide 3 cases. First a 0 case in which the slope is invariant in the y direction (main flow direction aligned with the grid). Second a 225 case, with the slope being invariant in a direction angled 225° from the y direction (main flow direction aligned

with the grid diagonal) and third a 120 case meaning that the slope is invariant in a direction angled 120° from the y direction (main flow direction is neither aligned with the grid nor with the grid diagonal). For each of these test cases, two simulations



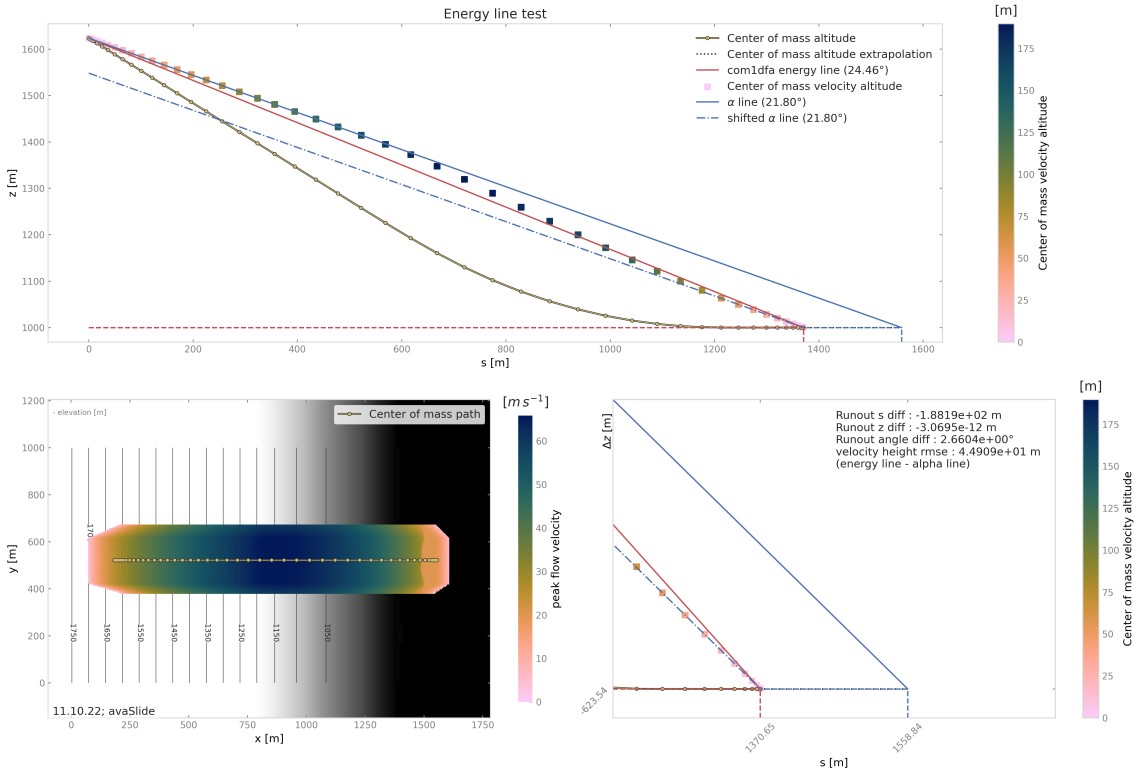

**Figure 13.** Energy line test for an inclined plane smoothly transitioning to a horizontal plane with particles following Coulomb friction, not subject to pressure forces but taking curvature acceleration into account. The geometrical solution and the numerical solution match on the inclined plane and horizontal parts and differ on the curved part. This shows the effect on curvature on the runout (decrease of runout because of the added friction due to curvature).

are performed, with or without pressure gradients. The results of these test and the instructions on how to reproduce them are provided in the supplementary material. The results are very satisfying. The runout distances differ from only a few centimeters between the 0°, 120° and 225° cases. This is of the same order of magnitude than the numerical errors computed with the energy line tests. This proves the rather low dependence of the simulation results on the grid orientation.

## 6 Conclusions

We present AvaFrame, an open source simulation tool for gravitational mass flows that aims to be used for operational hazard mapping as well as for research and education purposes. We give details of the theory background and numerical implementation of AvaFrame's computational dense flow module **com1DFA** at the date of publication of this manuscript. Additional information on the latest development steps will be included in real time in the online documentation of AvaFrame.





The **com1DFA** module is based on the slope normal thickness integration of the mass and momentum conservation equations which are solved with a mixed particle grid method. Particles are used to track mass and to compute the pressure forces following a smooth particle hydrodynamics (SPH) approach adapted to the special case of steep terrain. An underlying Eulerian grid is used to compute the flow thickness from the particles distribution. Velocity is also interpolated from the particles to the grid in order to compute the stabilizing artificial viscosity term. Physical starting and stopping behavior is ensured by taking into account differences in the friction force whether the material is flowing or at rest. A time space criterion ensuring the convergence of the implemented numerical method is provided.

We show the limitations of the assumptions, that are used to simplify the numerical resolution of the model equations. For example particles are forced to stay on the topography which means there is no free fall which can become important on strongly convex terrain features. This could be improved in future developments to allow the material to detach from the topography and enter a free fall state until it hits the topography again. This would lead to an improvement of model results in the case of a flow over a cliff or a bump.

The numerical implementation of **com1DFA** is verified by applying a series of tests. We separate these tests into two categories. Flow variable tests, i.e. similarity solution and dam break tests, are used for checking the proper spatio-temporal evolution of flow thickness and velocity. Runout testing is used for checking the accuracy of global variables such as center of mass runout and kinetic energy. The tests show the validity of the chosen time space criterion as well as the accuracy and precision of the **com1DFA** numerical solution. As part of the AvaFrame framework, the testing modules can be used independently from **com1DFA** and applied to other computational models aiming at solving the thickness integrated dense flow equations as well. The repertoire of tests used for verifying the numerical implementation of DFA models will be extended and the aim is to employ this test suite to continuously verify our **com1DFA** module throughout the developments.

Note that computational efficiency of the **com1DFA** module is not a topic in this manuscript since simulations with the standard setup compute within minutes on a 8 CPU core laptop. Of course increasing for example the spatial resolution, i.e. increasing number of particles and decreasing cell size, considerably increases the computation time (the limiting factor being related to the SPH pressure gradient step of the computation). Topics like further in depth testing and application to real topographies will be treated in future publications.

AvaFrame is an open-source framework and we invite everyone to make use of the presented modules and welcome feed back and contributions.

## Appendix A: Entrainment force

The mass of entrained snow for each particle depends on the type of entrainment involved (plowing or erosion) and reads:

$$\rho_0 \frac{\mathrm{d}}{\mathrm{d}t} \left( A_k^b \, h_k \right) = \frac{\mathrm{d}\, m_k}{\mathrm{d}t} = A_k^{\mathrm{ent}} \, q_k^{\mathrm{ent}} \tag{A1}$$



$A_k^{\text{ent}}$ respectively $q_k^{\text{ent}}$ represent the interface area between the particle and the entrainable material, respectively the mass flux. These quantities are expressed by:

$$A_k^{\text{plo}} = w_f\, h_k^{\text{ent}} = \sqrt{\frac{m_k}{\rho_0\, \overline{h}_k}}\, h_k^{\text{ent}} \quad \text{and} \quad q_k^{\text{plo}} = \rho_{\text{ent}}\, \|\overline{\mathbf{u}}_k\| \qquad \text{for plowing}$$

$$A_k^{\text{ero}} = A_k^b = \frac{m_k}{\rho_0\, \overline{h}_k} \quad \text{and} \quad q_k^{\text{ero}} = \frac{\tau_k^{(b)}}{e_b}\, \|\overline{\mathbf{u}}_k\| \qquad \text{for erosion} \tag{A2}$$

The term $\rho_0\, \overline{\mathbf{u}}_k\, \frac{\mathrm{d}(A^b\, h)}{\mathrm{d}t}$ related to the entrained mass in Eq. 30 accounts for the fact that the entrained material needs to be
accelerated from rest to the particle velocity. This term now reads:

$$\overline{\mathbf{u}}_k\, \rho_0\, \frac{\mathrm{d}}{\mathrm{d}t}\left(A_k^b\, h_k\right) = \overline{\mathbf{u}}_k\, A_k^{\text{ent}}\, q_k^{\text{ent}} \tag{A3}$$

## Appendix B: SPH pressure gradients

The SPH method used in shallow water equations is in most applications applied on a horizontal surface. The theoretical development on a horizontal plane is described in B1. The dense flow avalanche model described in this paper should be
expressed on the bottom surface which is not necessarily horizontal. The SPH gradient computation development is detailed in B2.

### B1   Standard method

Let us start with the computation of the gradient of a scalar function $f\colon \mathbb{R}^2 \to \mathbb{R}$ on a horizontal plane. Let $P_k = \mathbf{x}_k = (x_{k,1}, x_{k,2})$ and $Q_l = \mathbf{x}_l = (x_{l,1}, x_{l,2})$ be two points in $\mathbb{R}^2$ defined by their coordinates in the Cartesian coordinate system
$(P_k, \mathbf{e_1}, \mathbf{e_2})$. $\mathbf{r}_{kl} = \mathbf{x}_k - \mathbf{x}_l$ is the vector going from $Q_l$ to $P_k$ and $r_{kl} = \|\mathbf{r}_{kl}\|$ the length of this vector. Now consider the kernel function $W$:

$$W\colon \mathbb{R}^2 \times \mathbb{R}^2 \times \mathbb{R} \to \mathbb{R} \qquad\qquad , r_0 \in \mathbb{R} \text{ is the smoothing kernel length (or radius)} \tag{B1}$$
$$(P_k, Q_l, r_0) \mapsto W(P_k, Q_l, r_0)$$

In the case of the spiky kernel, $W$ reads (2D case):

$$W_{kl} = W(\mathbf{x_k}, \mathbf{x_l}, r_0) = W(\mathbf{x_k} - \mathbf{x_l}, r_0) = W(\mathbf{r_{kl}}, r_0)$$
$$= \frac{10}{\pi r_0^5} \begin{cases} (r_0 - \|\mathbf{r_{kl}}\|)^3, & 0 \le \|\mathbf{r_{kl}}\| \le r_0 \\ 0, & r_0 < \|\mathbf{r_{kl}}\| \end{cases} \tag{B2}$$

$\|\mathbf{r_{kl}}\| = \|\mathbf{x_k} - \mathbf{x_l}\|$ represents the distance between particle $k$ and $l$ and $r_0$ the smoothing length. Using the chain rule to express the gradient of $W$ in the Cartesian coordinate system $(x_1, x_2)$ leads to:

$$\boldsymbol{\nabla} W_{kl} = \frac{\partial W}{\partial r} \cdot \boldsymbol{\nabla} r, \quad r = \|\mathbf{r}\| = \sqrt{(x_{k,1} - x_{l,1})^2 + (x_{k,2} - x_{l,2})^2} \tag{B3}$$




with,

$$\frac{\partial W}{\partial r} = -3\frac{10}{\pi r_0^5}\begin{cases}(r_0 - \|\mathbf{r_{kl}}\|)^2, & 0 \leq \|\mathbf{r_{kl}}\| \leq r_0 \\ 0, & r_0 < \|\mathbf{r_{kl}}\|\end{cases} \tag{B4}$$

and

$$\frac{\partial r}{\partial x_{k,i}} = \frac{(x_{k,i} - x_{l,i})}{\sqrt{(x_{k,1} - x_{l,1})^2 + (x_{k,2} - x_{l,2})^2}}, \quad i = \{1,2\} \tag{B5}$$

which leads to the following expression for the gradient:

$$\boldsymbol{\nabla} W_{kl} = -3\frac{10}{\pi r_0^5}\begin{cases}(r_0 - \|\mathbf{r_{kl}}\|)^2\frac{\mathbf{r_{kl}}}{r_{kl}}, & 0 \leq \|\mathbf{r_{kl}}\| \leq r_0 \\ 0, & r_0 < \|\mathbf{r_{kl}}\|\end{cases} \tag{B6}$$

The gradient of $f$ is then simply:

$$\boldsymbol{\nabla} f_k = -\sum_l f_l A_l \,\boldsymbol{\nabla} W_{kl} \tag{B7}$$

**B2    2.5D SPH method**

We now want to express a function $f$ and its gradient on a potentially curved surface and express this gradient in the 3 dimensional Cartesian coordinate system $(P_k, \mathbf{e_1}, \mathbf{e_2}, \mathbf{e_3})$. Let us consider a smooth surface $\mathcal{S}$ and two points $P_k = \mathbf{x}_k = (x_{k,1}, x_{k,2}, x_{k,3})$ and $Q_l = \mathbf{x}_l = (x_{l,1}, x_{l,2}, x_{l,3})$ on $\mathcal{S}$. We can define $\mathcal{TP}$ the tangent plane to $\mathcal{S}$ in $P_k$. If $\mathbf{u}_k$ is the (none

zero) velocity of the particle at $P_k$, it is possible to define the local orthonormal coordinate system $(P_k, \mathbf{V_1}, \mathbf{V_2}, \mathbf{V_3} = \mathbf{n})$ with $\mathbf{V_1} = \frac{\mathbf{u}_k}{\|\mathbf{u}_k\|}$ and $\mathbf{n}$ the normal to $\mathcal{S}$ at $P_k$. Locally, $\mathcal{S}$ can be assimilated to $\mathcal{TP}$ and $Q_l$ to its projection $Q'_l$ on $\mathcal{TP}$ (see Fig. B1). The vector $\mathbf{r}'_{kl} = \mathbf{x}_k - \mathbf{x}'_l$ going from $Q'_l$ to $P_k$ lies in $\mathcal{TP}$ and can be express in the plane local basis:

$$\mathbf{r}'_{kl} = \mathbf{x}_k - \mathbf{x}'_l = v_{kl,1}\mathbf{V_1} + v_{kl,2}\mathbf{V_2} \tag{B8}$$

It is important to define $f$ properly and the gradient that will be calculated:

$$\begin{aligned}f : \mathcal{TP} \subset \mathbb{R}^3 &\to \mathbb{R} \\ (x_1, x_2, x_3) &\mapsto f(x_1, x_2, x_3) = f(x_1(v_1, v_2), x_2(v_1, v_2)) = \widetilde{f}(v_1, v_2)\end{aligned} \tag{B9}$$

Indeed, since $(x_1, x_2, x_3)$ lies in $\mathcal{TP}$, $x_3$ is not independent of $(x_1, x_2)$:

$$\begin{aligned}\widetilde{f} : \mathcal{TP} \subset \mathbb{R}^2 &\to \mathbb{R} \\ (v_1, v_2) &\mapsto \widetilde{f}(v_1, v_2) = \widetilde{f}(v_1(x_1, x_2), v_2(x_1, x_2)) = f(x_1, x_2, x_3)\end{aligned} \tag{B10}$$

The target is the gradient of $\widetilde{f}$ in terms of the $\mathcal{TP}$ variables $(v_1, v_2)$. Let us call this gradient $\boldsymbol{\nabla}_{\mathcal{TP}}$. It is then possible to apply the B1 method to compute this gradient:

$$\boldsymbol{\nabla}_{\mathcal{TP}} W_{kl} = \frac{\partial W}{\partial r} \cdot \boldsymbol{\nabla}_{\mathcal{TP}} r, \quad r = \|\mathbf{r}\| = \sqrt{v_{kl,1}^2 + v_{kl,2}^2} \tag{B11}$$





**Figure B1.** Tangent plane and local coordinate system used to apply the SPH method

Which leads to:

$$\boldsymbol{\nabla}_{\mathcal{TP}}W_{kl} = -3\frac{10}{\pi r_0^5}\frac{(r_0 - \|\mathbf{r'_{kl}}\|)^2}{r'_{kl}}\begin{cases} v_{kl,1}\mathbf{V_1} + v_{kl,2}\mathbf{V_2}, & 0 \le \|\mathbf{r'_{kl}}\| \le r_0 \\ 0, & r_0 < \|\mathbf{r'_{kl}}\| \end{cases} \tag{B12}$$

$$\boldsymbol{\nabla}_{\mathcal{TP}}\widetilde{f}_k = -\sum_l \widetilde{f}_l A_l \,\boldsymbol{\nabla}_{\mathcal{TP}}W_{kl} \tag{B13}$$



This gradient can now be expressed in the Cartesian coordinate system. It is clear that the change of coordinate system was not needed:

$$\boldsymbol{\nabla}_{\mathcal{TP}} W_{kl} = -3 \frac{10}{\pi r_0^5} \frac{(r_0 - \|\mathbf{r'_{kl}}\|)^2}{r'_{kl}} \begin{cases} r_{kl,1}\mathbf{e_1} + r_{kl,2}\mathbf{e_2} + r_{kl,3}\mathbf{e_3}, & 0 \leq \|\mathbf{r'_{kl}}\| \leq r_0 \\ 0, & r_0 < \|\mathbf{r'_{kl}}\| \end{cases} \quad\quad \text{(B14)}$$

The advantage of computing the gradient in the local coordinate system is if the components (in flow direction or in cross flow direction) need to be treated differently.

*Code and data availability.* The AvaFrame software is publicly available at https://github.com/avaframe/AvaFrame/ and https://doi.org/10.5281/zenodo.6806813 (Oesterle et al., 2022). The associated online documentation is available at https://docs.avaframe.org/en/latest/. The code is released under the European Union Public license (EUPL version 1.2). The version of the code (including configuration files) and documentation for reproducing the results presented in this paper are available at https://github.com/avaframe/AvaFrame/tree/theoryPaperCode and https://docs.avaframe.org/en/theorypapercode/index.html

*Author contributions.* AW, FO and MT are the principal programmers and jointly developed the Dense Flow Avalanche numerical model code. MT prepared the manuscript and performed the model simulations to produce the results and figures of this manuscript with contributions from AW. F0 and JTF jointly supervised the project and contributed to the discussion and paper.

*Competing interests.* The contact author has declared that none of the authors has any competing interests.

*Acknowledgements.* The authors would like to acknowledge Peter Sampl who developed the theory and source code for SamosAT which
represented the starting point of the **com1DFA** module of AvaFrame. Thank you to Marie von Busse and Wolfgang Fellin for their help in developing the energy line test.

*Financial support.* AvaFrame is supported by the Austrian Federal Ministry for Agriculture, Forestry, Regions and Water Management through a cooperation between the Austrian Avalanche and Torrent Control (WLV) and the Austrian Research Centre for Forests (BFW). Additional support is provided by the international cooperation project "AvaRange - Particle Tracking in Snow Avalanches" supported by
the German Research Foundation (DFG, project No. 421446512) and the Austrian Science Fund (FWF, project No. I 4274-N29).



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
