# Peer review of "AvaFrame com1DFA (version 1.3): a thickness-integrated computational avalanche module - Theory, numerics and testing"

_EGUsphere, 2022_

## Referee Comment (RC2)

        2021.

[referee-annotated manuscript omitted]

---

## Author Comment (AC1)

Dear Reviewer 1,

Thank you for taking the time to review our submission, this is very much appreciated. You will find replies to your comments and issues below. We hope we addressed your issues to your liking. We copied the comments with a specific reply to the top of this reply and collected the ones we simply addressed at the bottom.

L64: It is not clear to me how the word "AvaFrame" fits in this sentence. Further, the "F"in "AvaFrame" is written in upper case here and in other places in the manuscript, but in lower case in the title of the paper. Please ensure consistency, or explain why there is this difference.

-> We removed the word AvaFrame here. Furthermore we changed the title to be consistent with the rest of the paper.

L71: Maybe I overlooked it - did you introduce the abbreviation "DFA" before? It is very clear what it means, but still it has to be defined.

-> No, you did not overlook it, it was missing. Added the definition at the first occurrence of DFA in the introduction.

L437: Is this really a dam break? I would rather describe it as the collapse of a cliff. Shouldn't a dam break imply that there is some fluid behind, which is released?

-> We use the same designation as the literature we are referencing. Furthermore we are handling a fluid, whereas a cliff would imply breaking/fracture forces, which we do not include. Maybe the issue is the missing depiction of the dam? We basically start with an (admittedly artificial) wall of fluid at t=0.

The following remarks we simply addressed:

L23f: "... combining over a decade of operational application ...": This formulation is not very clear, maybe better mention that this operational application refers to SamosAT?

L42: "... proposed in Mangeney-Castelnau et al. (2003), ..."

L144f: Please check brackets in the reference to the Voellmy model.

L255: I am not a native speaker, but "... using such an SPH method ..." might be better.

L256: "... previously described grid method ..."

L311: "... used in the com1DFA code ..."

L321f: Shouldn't the DEM also be described by the number of cells in x and y direction?

L400: Maybe better: "... a (semi-)analytical solution ...". But I am not sure, it is a tricky issue.

L411: The similarity solution and dam break tests come very sudden here (not

mentioned before). Maybe better reformulate this sentence a little bit. L474: "... between analytical and numerical solution ..." L480: "... with an α exponent ..." L481f: "... α = 0 for the dam break test ..." Figure 4, header: "Similarity solution test ..." L493: "... only Coulomb friction ..." L541: "... and questions like ..." L562: "The results of these tests ..." L569: "at the date of publication of this manuscript": this formulation sounds somehow strange to me, better reformulate. L590: Better: "... throughout the development." L591: "Note that the computational efficiency ..."

L596f: "feed back" --> "feedback"

\_\_\_\_\_

Dear Dieter,

Thank you for taking your time to review our submission. Your thorough comments are very much appreciated and highly valued. Please find our replies below.

The paper is clearly structured, and with a few exceptions indicated below (Minor remarks) or marked in the manuscript, all points are suitably explained. At a few places, the mathematical derivations are on the explicit side and could be shortened. In a large part of the manuscript, the English is good. There are, however, passages with grammar errors and at times rather clumsy and even ambiguous formulations. The annotations contain some suggestions on how to improve this.

Nevertheless, the authors need to critically re-read the text and bring these passages up to the level of the rest.

-> We reworked quite a few sections (see specific comments below) and addressed the annotations.

It could be useful for the readers if the authors cited the most relevant open-source models for geophysical mass flows, like Titan2D, Clawpack and its derivatives or MassMov2D, and perhaps summarized the salient differences from their approach (from a user perspective).

-> While we see the benefit of including a deeper discussion about the differences to other popular models, we did not include this in the current manuscript. We are currently also working on another manuscript in which we will give a broader overview over the AvaFrame framework (vs the focus on com1DFA in the current one). There we will include such a comparison. We think the current submission already contains quite a lot of information.

**Major remarks**

To the best of my understanding, the geometric effects are rendered correctly in the formulas. However, the explanations given in the text appear confusing if not misleading, particularly in Appendix B2.

It is crucial to distinguish between the terrain surface, which is a 2D manifold S embedded in 3D Euclidian space E3, and the 2D Euclidian tangent spaces Tx, one associated with each point x on the manifold at a given time. In some contexts, these 2D tangent spaces are extended to 3D (Euclidian) spaces Tx3 in the manuscript; this can be done naturally, but these spaces are not the same conceptually as the embedding space E3. The fields h, u (and also the stress tensor  $\sigma$ ) are not functions from the tangent plane, as stated in Appendix B2, to Euclidian space or R but from the surface S to the non-negative real numbers (h), the Tx or Tx3 (u) and a 9D space for the stress tensor. The appropriate modern mathematical concept to use would be the fiber bundle. I think that the fiber bundles in the present situation are essentially trivial, provided the fibers for the field h are limited to the interval [0,1/kmax[, where kmax is the maximum of curvature (in any direction) of the base manifold S (the topography). The same results can, however, also be obtained with classical differential geometry. A fundamental result is that the whole theory can be developed without reference to the Euclidian space E in which S is embedded, but the price to pay is using the metric tensor and its derivatives, from which the connection coefficients (Christoffel symbols) can be derived. Most models known to me that tackle at least some of the intricacies of flows over curved surfaces use this approach, defining a coordinate system on S and letting the tangent space at a given point be spanned by the tangent vectors to the coordinate lines through that point. These coordinate systems cannot be orthonormal unless S is flat.

In contrast to this, the authors explicitly use the embedding Euclidian space E3, which allows to give absolute orientation to all tangent spaces (and their 3D extensions) associated with the points on S. The "natural coordinate system" (NCS) used by the authors is not a coordinate system on S or in E3 but a separate coordinate system in each of the Tx3; moreover, the NCSs are time-dependent, being advected along with the "particles". The choice of spatially and temporally varying coordinate systems in each of the extended tangent planes would be a computational nightmare. However, the authors cleverly exploit the fact that, in essence, only the basis vector v1(x,t) is needed to compute the gradient of the kernel W.

To be clear: I do not criticize the authors' choice of dealing with the geometry—it does indeed have several attractive properties. However, for the readers' sake I wish they clearly distinguished between base space and target space of the functions and briefly explained the difference with traditional approaches.

-> To address your points, we reworked the beginning of the NCS section. Furthermore we tried to address this throughout the whole manuscript and add the necessary information both in text and in equations. We hope we are now clearer in distinguishing between the different coordinate systems.

There are also some points that are important for understanding the method but are not explained as far as I can see. In the mass-balance equation (1),  $dV(x,t)/dt = q(x,t)/\rho dA$ , and Eq. (5) implies dV = h(x,t) dA. This shows that dV/dt = dh/dt dA + h dA/dt, i.e., in order to find the flow depth one must know how the basal area of the SPH particle changes over time. To this end, one will need the 2D divergence of the velocity field. I think I can see how this can be obtained by expressing the basis vectors at all points in terms of the global Cartesian coordinate system, but I cannot find this explained in the manuscript.

-> We handle this really simplified (basically A = m/(rho h)), and we point to equation (A2) in the original submission. We only need A for entrainment and special friction laws (eg. SamosAT/Chezy) where A might be needed. But we also see your point and this might be a solution worth investigating/exploring further.

In this context, I am also surprised that the terrain curvature (more precisely, the Gaussian curvature) only briefly appears in a formula after Eq. (4), even though it plays a significant role in the equation of motion later on, where it is masked as dv3/dt. I think the mathematical development would gain clarity if the connection between dv3/dt,  $\nabla$ v3 and curvature  $\kappa$  were stated explicitly. Given there are many good Eulerian mesh-based solvers for hyperbolic equations and that com1DFA seems to be fairly complex and not in the top league when it comes to speed, the authors ought to explain why they chose this approach.

-> We actually do not need to use Kappa, as with our choice of coordinate system, the curvature term is hidden in the time derivative of the Normal vector to the surface S (which we compute directly). We included a short info about this in the text.

**Moreover, it is**

known that boundary conditions are not straightforward to implement in SPH codes, thus at least a brief discussion of how this is done here (or avoided if the particles' kernel areas never approach the domain boundary) should be included.

-> com1DFA does not include boundaries, we make sure the kernel areas never approach the domain boundary. Particles are removed if they reach the end of the domain. There is no way to include impenetrable boundaries (also see next reply).

There is one test of the model I would like to see, namely impact on a wall or, if the code does not implement impenetrable boundaries, run-up on a steep counter-slope. Such a situation occurs in many practical applications. This serves to test the shock-capturing capabilities of the code, as it has to create a backward traveling shock. The simplest form of this test is for an inviscid jet traveling horizontally with a prescribed speed and flow depth before hitting a vertical wall. In this case, the shock speed and height can be obtained analytically.

-> We never thoroughly checked how com1DFA would react to discontinuities in the slope normal vector (wall). And we see no reason why it should work since we did not plan it, as it is a clear violation of the "moderately curved surface" assumption. So we would need to add some tricks in the code to deal with this phenomenon. However, based on your suggestion we intend to include a shock-capturing test in our set of standard-tests we provide with the framework (see the issue at github https://github.com/avaframe/AvaFrame/issues/858). We already include a 'wall approximation (wall lines)' in our code, in which the user can specify lines which act like walls. This might be useful as a simple/quick tool to investigate effects of directional dams. However these are solved with a workaround by simply slowing down and redirecting the particles artificially. Again, this is only intended as a first evaluation function, not as a way to directly/realistically include wall and catching dams. This is definitely an area for future improvement.

In a test with friction on a slope, the shape of the final deposit would also be of interest because many practically used models of Voellmy-type (of which the Coulomb model presented here is a special case) tend to creep (sometimes intermittently) for very long times until the surface slope of the deposit nowhere exceeds the bed friction angle.

-> We don't model compaction or cohesion, so this behaviour is expected. It will stop at one point. To address a similar effect, we implemented a explicit friction, but this is in an experimental option and not tested yet. See our documentation at https://docs.avaframe.org/en/latest/DFAnumerics.html#an-explicit-method for more information.

Section 5.2 (except Sec. 5.2.3) takes a rather large fraction of the entire manuscript, even though the theory behind these tests has in its essence been known for 40 years or perhaps even for a century. Given the results of the tests presented in Sec. 5.1, I would have been surprised if the energy-line tests had shown poor results.

-> While we are also not surprised at the good results, we are actually surprised that the results are this good. The test is interesting because it double-checks the correct velocity evolution along the path and a correct stopping behaviour of the code - without imposing any "artificial" stopping criteria. We agree that this is not surprising considering the results of the flow variable test, however the flow variable testing does not operate on the large scale (which is important for operational purposes). The energyline test on the other hand could be easily applied to any model of the same class without performing any flow variable testing. But to address your concerns about the length, we moved the theory section to the appendix and shortened the 'limitations and applicability' section (and kept the grid orientation section as per your recommendation).

**This part could easily be**

shortened. Section 6 is titled "Conclusions", but except for lines 591-595 it is simply a brief summary of the preceding five sections. Readers who have read the paper, will be bored, and those who just read the abstract and skipped the rest will not be enlightened either. For this section to be interesting, the authors could try to answer some of the following questions: 1) For which type of application is com1DFA particularly suitable, for which is it not suitable (at least presently)? 2) How does com1DFA compare to other codes "on the market"? 3) What are the most interesting, useful and feasible extensions of com1DFA?

-> We rewrote the conclusions, trying to address your questions and make it more interesting to the 'full paper' readers. However, since this manuscript aims to present our model and how we do it, we did not include a comparison to other codes. We will tackle this in a future publication where we present the whole AvaFrame framework. Especially since there we will also highlight the different computational modules with the different concepts behind them. com1DFA is only one approach of the already implemented ones in AvaFrame.

Lines 591-595 contain relevant information with regard to practical applications. The authors state that "simulations compute ... within minutes on an 8-core laptop." This suggests that the code uses some degree of parallel processing (for which SPH is well suited). From my own and my colleagues' experience, I consider this to be a bottleneck because consultants often run perhaps ten simulations per path to test sensitivity to the choice of parameters and hate to wait for an hour to get the results. Other codes like NGI's MoT-Voellmy achieve such simulations on a single core typically within seconds to tens of seconds (admittedly at less precision, but this is of little concern if the initial conditions are as poorly known as usually is the case).

-> We realize this was poorly formulated and open to misinterpretation. We reformulated it. To make it clear: a single com1DFA simulation itself is not parallelized! The combination of python and cython makes it fast enough to not warant a full parallelization via OPENMP/MPI or similar (yet). The low-hanging fruit, which you already observed, is the parallel run of multiple simulations. The speedups one achieves vs the effort spent on implementing it is substantially better than the OPENMP/MPI route. We already implemented parallel simulations in version 1.5. We also included a very simple way to run a range of parameter values, specifically with sensitivity and range tests in mind. (See the last paragraph of this section: https://docs.avaframe.org/en/latest/advancedUsage.html#configuration)

In Appendix A, the authors state the mathematical formulation of the entraiment models that have been implemented in com1DFA, even though they are not used in this manuscript. This is valuable as a documentation of the present state of the code. For this reason, a few critical remarks concerning these formulas may be in order:

In Eq. (A2), neither pent nor eb are defined. The expression  $(mk/(\rho 0hk))1/2$  seems to be used as the width of the SPH particle. This would seem to apply only to a square-shaped particle.

-> Yes, that is true.

**The formula for the entrainment rate due**

to plowing uses the flow velocity. It has been pointed out already in the 1960s (Eglit, Grigorian, Yakimov) that the entrainment front propagates faster than the flow. It would seem more physically justifiable to create new particles from the plowed snow cover, with the rate of particle creation accounting for the shock propagation velocity whereas the new partcles themselves move at the lower speed of the flow front. Another weakness of the proposed formula is that the erosion depth hent nust be specified beforehand; on physical grounds, it should be determined dynamically. The formula does not account for the shear/compression resistance of the snow cover either (see (Eglit et al., 2020) for more details). The formula for scouring erosion has similar shortcomings in that it does not account for the shear strength of the snow-unless this is somehow included in eb (if so, this needs to be explained!). I do not understand the physical reasoning behind the factor ||u||-based on the arguments in (Issler and Pastor, 2011) or (Issler, 2014) I would expect 1/u.

-> Yes, we acknowledge the shortcomings of our entrainment approach with various disadvantages. At its current state it is a 'workhorse' approach, i.e. it is not very sophisticated, but gets the work done for our main (operational) application. As it is not necessary in the context of this paper, we decided to remove this section completely and leave this open for further discussions on possible improvements. However, we also see this as an area where improvement would be very beneficial. As AvaFrame is open-source, feel free to add your knowledge to it. Maybe we find a way to build on your publications and include your methods in future releases.

**Minor remarks**

In the attached commented manuscript, there are about two-hundred remarks that the authors should look at when revising the manuscript. I would, however, ask them not to reply individually to each of them, irrespective of what the journal's policies are. Instead, please comment only on the most important remarks you do not agree with! The vast majority of the remarks concerns language issues.

**----**

Remarks regarding the annotations and minor remarks:

-> the 'thickness' issue: while we acknowledge the fact that depth/height is being used in many previous publication, we also point out that these terms are quite often used inconsistently (not to speak about usage in other, related fields of research; meteorology, glaciology, etc)! To avoid any confusion, we decided to go with thickness, which is clearly defined: "the smallest of three dimensions" (webster). E.g, in the context of a wooden board, the shortest distance between two surfaces. This removes any ambiguity about surface normal vs gravity direction. Therefore we think it is of benefit for our communication and will stick with it. We also think it would benefit the avalanche modelling community in general, but only the future will show whether it will actually catch on.

-> Regarding the citet vs citep issue: we went with the GMD style guide, see https://www.geoscientific-model-development.net/submission.html . This means whenever there is the textual citation '... is well described in Liu and Liu (2010)', only the year is in parentheses.

-> Regarding the annotations about the math style elements we used: we consulted the editor and decided to keep both the crossouts as well as leave the superscripts without parentheses. However, we went with a suggestion by the editor and increased spacing around the crossouts.
* * *
The authors state that they use prismatic control volumes (or particles). This would lead to "crevasses" in the flow body wherever the terrain curvture is negative, and to "caves" (or, alternatively, interpenetration of parts of the control volumes) in areas of positive cutrvature. Perhaps they mean truncated pyramidal volumes instead? The description of the SPH method is rather confusing in my eyes, as it is being said that the value of some quantity at the location of particle j is the weighted sum of that quantity over all other particles. But the same quantity at the location of another particle depends, among others, on the quantity of the first particle. In the extreme case with only 2 particles, we get f1 = w12f2 and f2 = w21f1 = w12f1 = w122f1  $\Rightarrow$  w12 = ±1  $\Rightarrow$  f2 = ±f1,

which is hardly what is intended. The way I understand SPH is that it provides a way of interpolating a field value at any location from the corresponding quantity associated with a finite number of locations (i.e., particles). This seems to be supported, e.g., by the documentation of Abaqus, which states:

"At its core, the method is not based on discrete particles (spheres) colliding with each other in compression or exhibiting cohesive-like behavior in tension as the word particle might suggest. Rather, it is simply a clever discretization method of continuum partial differential equations. In that respect, smoothed particle hydrodynamics is quite similar to the finite element method. SPH uses an evolving interpolation scheme to approximate a field variable at any point in a domain. The value of a variable at a particle of interest can be approximated by summing the contributions from a set of neighboring particles, denoted by subscript j, for which the "kernel" function, W, is not zero:  $(f(x)) \approx \sum j$  mj pj fj W(||x-xj||,h)."

-> Yes, it is problematic when one needs f to compute f. But there is a trick. We want to compute the flow thickness h so f=h. And Aj\*hj= massj/rho

 $(h(x)) \approx \sum j$  hj W(||x-xj||,rKernel) Aj =  $\sum j$  mj/rho W(||x-xj||,rKernel) we know mj since it is the mass of the particle which is defined in the initialisation and does not change (or changes if there is entrainment but we know how it changes). So we do not have the problem from needing h to compute h. We need m to compute h and we know m.

I am not very familiar with the SPH method, but I think other readers might stumble over the same issue; therefore, I ask the authors to consider this naïve objection and to make the description as intuitive as possible for a non-specialist audience. In particular, it would be important to explain at which stage this averaging over neighboring particles is carried out—I could not find any mentioning of it in the text.

-> We explicitly included a new equation and reference it in the section driving forces

The authors should explain why they think that the convergence criteria developed by Moussa & Vila (2000) also apply in the case at hand.

-> We made the connection more obvious in text: i.e. it being the same type of equations with the same methods.

I think that the dam-break solution (Fig. 6) can be applied at the upstream end of the released mass as well (the solution consists of a dam break towards the left on a horizontal plane with gravity reduced by a factor  $\cos \theta$ , superposed with a uniformly accelerated downstream motion). Doing so would provide yet another test of the code.

-> We did not consider this yet, only concentrating on the downstream side. We will investigate this for future testing.